# Fast-dLLM: Training-free Acceleration of Diffusion LLM by Enabling KV Cache and Parallel Decoding

**Chengyue Wu**[1,2]* **Hao Zhang**[2]* **Shuchen Xue**[2] **Zhijian Liu**[2] **Shizhe Diao**[2] **Ligeng Zhu**[2]
**Ping Luo**[1] **Song Han**[2,3] **Enze Xie**[2]
[1]The University of Hong Kong   [2]NVIDIA   [3]MIT

*Equal contribution.
https://nvlabs.github.io/Fast-dLLM

## ABSTRACT

Diffusion-based large language models (Diffusion LLMs) have shown promise for non-autoregressive text generation. However, the practical inference speed of open-sourced Diffusion LLMs often lags behind autoregressive models due to the lack of Key-Value (KV) Cache and quality degradation when decoding multiple tokens simultaneously. To bridge this gap, we introduce Fast-dLLM, a method that incorporates a novel block-wise approximate KV Cache mechanism tailored for bidirectional diffusion models, enabling cache reuse with negligible performance drop. Additionally, we identify the root cause of generation quality degradation in parallel decoding as the disruption of token dependencies under the conditional independence assumption. To address this, Fast-dLLM also proposes a confidence-aware parallel decoding strategy that selectively decodes tokens exceeding a confidence threshold, mitigating dependency violations and maintaining generation quality. Experimental results on LLaDA and Dream models across multiple LLM benchmarks demonstrate up to 27.6× throughput improvement with minimal accuracy loss, closing the performance gap with autoregressive models and paving the way for practical deployment of Diffusion LLMs.

## 1 INTRODUCTION

Diffusion-based large language models (Diffusion LLMs) have recently attracted increasing attention due to their potential for parallel token generation and the advantages of bidirectional attention mechanisms. Notably, Mercury (Inception Labs, 2025) runs at over 1,000 tokens per second, and Gemini Diffusion (Google DeepMind, 2025) by Google DeepMind has demonstrated the ability to generate over 1,400 tokens per second, highlighting the promise of significant inference acceleration.

However, current open-source Diffusion LLMs (Nie et al., 2025b; Ye et al., 2025) have yet to close such throughput gap in practice, and their actual speed often falls short of autoregressive (AR) models. This is primarily due to two issues. First, diffusion LLMs do not support key-value (KV) caching, a critical component in AR models for speeding up inference. Second, the generation quality tends to degrade when decoding multiple tokens in parallel. For example, recent findings such as those from LLaDA (Nie et al., 2025b) indicate that Diffusion LLMs perform best when generating tokens one at a time and soon degrades when decoding multiple tokens simultaneously.

To bridge the performance gap with AR models that benefit from KV Cache, we present Fast-dLLM, a fast aWnd practical diffusion-based language modeling framework. First, Fast-dLLM introduces an approximate KV Cache tailored to Diffusion LLMs. While the bidirectional nature of attention in Diffusion LLMs precludes a fully equivalent KV Cache, our approximation closely resembles an ideal cache in practice. To support KV Cache, we adopt a block-wise generation manner. Before generating a block, we compute and store KV Cache of the other blocks to reuse. After generating the block, we recompute the KV Cache of all the blocks. Visualizations confirm the high similarity with adjacent inference steps within the block, and our experiments show that this approximation

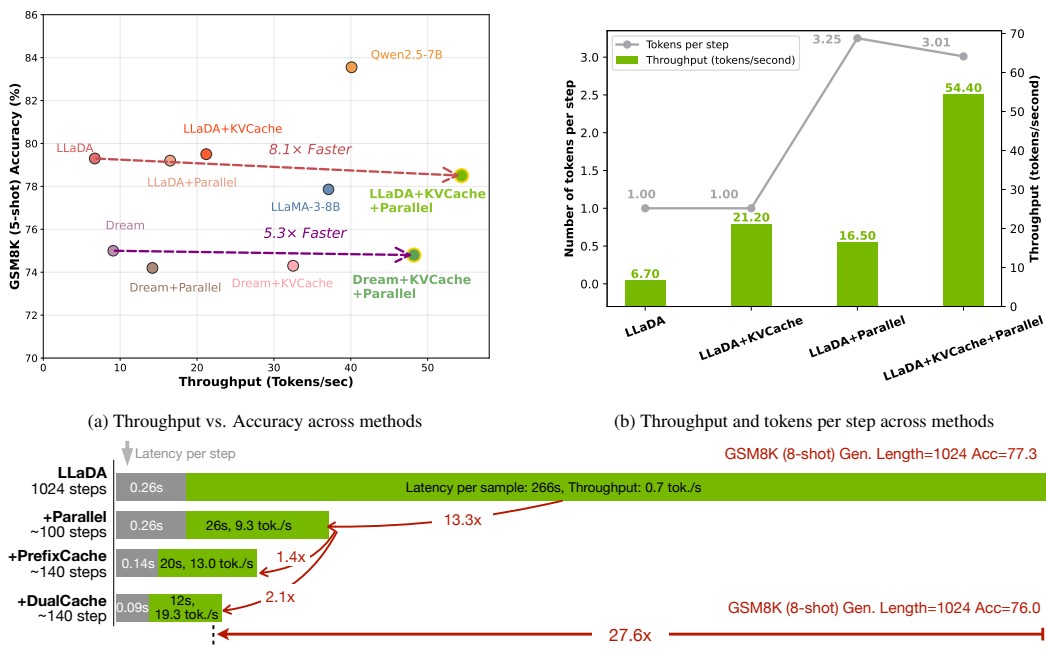

(a) Throughput vs. Accuracy across methods

(b) Throughput and tokens per step across methods

(c) End-to-end speedup over vanilla LLaDA baseline.

Figure 1: **Effectiveness of components of Fast-dLLM across different approaches.** We use NVIDIA A100 GPU with a single batch size and no inference speedup frameworks.. **(a)** Inference throughput (tokens/sec) and GSM8K (5-shot) accuracy across various designs and models under a maximum generation length of 256. Caching mechanism and parallel decoding can significantly accelerate inference, while the combination provides up to an 8.1× increase in throughput with negligible accuracy reduction. **(b)** We break down the contributions of each method by showing both the number of tokens generated per step (line) and total throughput (bars). **(c)** With long prefilling (8-shot) and a maximum generation length of 1024, our combined approach achieves up to 27.6× end-to-end speedup compared to the vanilla LLaDA baseline.

preserves model performance during inference. We further propose a DualCache version that caches Keys and Values for both prefix and suffix tokens.

In parallel, Fast-dLLM investigates the degradation in output quality when generating multiple tokens simultaneously. Through theoretical analysis and empirical studies, we identify that simultaneous sampling of interdependent tokens under a conditional independence assumption disrupts critical token dependencies. To address this issue and fully exploit the parallelism potential of Diffusion LLMs, we propose a novel confidence-thresholding strategy to select which tokens can be safely decoded simultaneously. Instead of selecting the tokens with top K confidence to decode as in LLaDA, we select tokens with confidence larger than a threshold. Our theoretical justification and experimental results demonstrate that this strategy maintains generation quality while achieving up to 13.3× inference speed-up.

In summary, our contributions are threefold. First, **Key-Value Cache for Block-Wise Decoding.** We introduce a block-wise approximate KV Cache mechanism specifically designed for bidirectional attention. Our approach reuses cached activations from previously decoded blocks by exploiting the high similarity of KV activations between adjacent steps. By caching both prefix and suffix blocks, the DualCache strategy enables substantial computational reuse. Second, **Confidence-Aware Parallel Decoding.** We propose a novel confidence-aware parallel decoding method. Unlike prior approaches that select a fixed number of tokens per step, our method dynamically selects tokens whose confidence exceeds a global threshold, enabling safe and effective parallel decoding. This approach significantly accelerates inference by 13.3× while preserving output quality. Third, **State-of-the-Art Acceleration Results.** We conduct comprehensive experiments on multiple open-source Diffusion LLMs (LLaDA, Dream) and four mainstream benchmarks (GSM8K, MATH, HumanEval, MBPP). Results demonstrate that our Fast-dLLM consistently deliver order-of-magnitude speedups with minimal or no degradation in accuracy, confirming the generality and practical value of our

approach for real-world deployment. Fast-dLLM achieves hgiher acceleration (up to 27.6×) when generation length is longer (1024).

## 2 PRELIMINARY

### 2.1 MASKED DIFFUSION MODEL

Diffusion models for discrete data were first explored in (Sohl-Dickstein et al., 2015; Hoogeboom et al., 2021). D3PM (Austin et al., 2021) generalized them with a discrete-state Markov chain forward process parameterized by transition matrices $\boldsymbol{Q}_t$, and learned the reverse process $p_\theta(\boldsymbol{x}_0|\boldsymbol{x}_t)$ via ELBO maximization. CTMC (Campbell et al., 2022) extended this to continuous time, while SEDD (Lou et al., 2023) instead modeled the likelihood ratio $\frac{p_t(\boldsymbol{y})}{p_t(\boldsymbol{x})}$ using Denoising Score Entropy.

Among noise processes, Masked Diffusion Models (MDMs)—also called absorbing state discrete diffusion—are prominent. MDMs replace tokens with a special [MASK] token according to

$$q_{t|0}(\boldsymbol{x}_t|\boldsymbol{x}_0) = \prod_{i=1}^{n} \mathrm{Cat}\Big(\boldsymbol{x}_t^i; (1-t)\delta_{\boldsymbol{x}_0^i} + t\delta_{[\mathrm{MASK}]}\Big), \tag{1}$$

where $t \in [0, 1]$ interpolates between $\boldsymbol{x}_0$ ($t = 0$) and a fully masked sequence ($t = 1$).

Recent work (Shi et al., 2024; Sahoo et al., 2024; Zheng et al., 2024; Ou et al., 2024) shows MDM parameterizations are equivalent and that their training objective reduces to an ELBO:

$$-\log p_{\boldsymbol{\theta}}(x) \leq \int_0^1 \frac{1}{t} \mathbb{E}_{q_{t|0}}\Bigg[\sum_{i:\boldsymbol{x}_0^i=[\mathrm{MASK}]} -\log p_{\boldsymbol{\theta}}(\boldsymbol{x}_0^i|\boldsymbol{x}_t)\Bigg]\mathrm{d}t := \mathcal{L}_{\mathrm{MDM}}. \tag{2}$$

### 2.2 GENERATION PROCESS OF MDMS

Directly reversing Equation 1 is inefficient, altering only one token per step (Campbell et al., 2022; Lou et al., 2023). A faster strategy is $\tau$-leaping (Gillespie, 2001), which lets multiple masked tokens be recovered in a single step from $t$ to $s < t$:

$$q_{s|t} = \prod_{i=0}^{n-1} q_{s|t}(\boldsymbol{x}_s^i|\boldsymbol{x}_t), \ q_{s|t}(\boldsymbol{x}_s^i|\boldsymbol{x}_t) = \begin{cases} 1, & \boldsymbol{x}_t^i \neq [\mathrm{MASK}], \boldsymbol{x}_s^i = \boldsymbol{x}_t^i \\ \frac{s}{t}, & \boldsymbol{x}_t^i = [\mathrm{MASK}], \boldsymbol{x}_s^i = [\mathrm{MASK}] \\ \frac{t-s}{t}q_{0|t}(\boldsymbol{x}_s^i|\boldsymbol{x}_t), & \boldsymbol{x}_t^i = [\mathrm{MASK}], \boldsymbol{x}_s^i \neq [\mathrm{MASK}]. \end{cases} \tag{3}$$

Here $q_{0|t}(\boldsymbol{x}_s^i|\boldsymbol{x}_t)$ is a model distribution over the vocabulary, extended to $q_{0|t}(\boldsymbol{x}_s^i|\boldsymbol{x}_t, p)$ when conditioned on a prompt $p$.

**Curse of Parallel Decoding** Although $\tau$-leaping accelerates generation by sampling multiple tokens in parallel, the conditional independence assumption causes problems. For example, in "The list of poker hands that consist of two English words are: _ _" (Song & Zhou, 2025), valid pairs include "high card" or "full house," but independent sampling can yield incoherent pairs like "high house." Formally, MDMs approximate $p(\boldsymbol{x}_s^i, \boldsymbol{x}_s^j|\boldsymbol{x}_t)$ by $p(\boldsymbol{x}_s^i|\boldsymbol{x}_t)\,p(\boldsymbol{x}_s^j|\boldsymbol{x}_t)$, ignoring dependencies such as $p(\boldsymbol{x}_s^j|\boldsymbol{x}_t, \boldsymbol{x}_s^i)$. This mismatch worsens when many tokens are unmasked simultaneously, degrading fluency and coherence.

## 3 METHODOLOGY

### 3.1 PIPELINE OVERVIEW

Our approach, Fast-dLLM, builds on the Masked Diffusion Model (MDM) architecture to enable efficient and high-quality sequence generation. To accelerate inference, the overall pipeline incorporates two key strategies: efficient attention computation through Key-Value (KV) Cache and a parallel decoding scheme guided by prediction confidence.

Specifically, we adopt Key-Value Cache for Block-Wise Decoding, which allows reusing attention activations across steps and significantly reduces redundant computation. Within each block, we

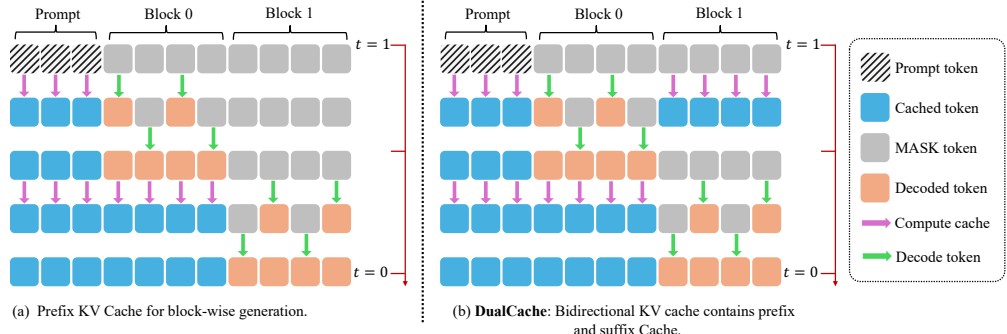

Figure 2: **Illustration of our Key-Value Cache for Block-Wise Decoding.** (a) During prefix-only caching, the KV cache is computed once for the prompt and reused across multiple decoding steps within each block. The cache is updated after completing a block to maintain consistency, with negligible overhead. (b) DualCache extends this approach by caching both prefix and masked suffix tokens, further accelerating decoding. The high similarity of KV activations across steps allows effective reuse with minimal approximation error.

further propose Confidence-Aware Parallel Decoding, enabling selective updates of tokens based on confidence scores to improve efficiency while maintaining output quality.

By combining these strategies, Fast-dLLM significantly speeds up inference for MDMs with minimal impact on generation performance. The overall procedure is summarized in Algorithm 1.

## 3.2 Key-Value Cache for Block-Wise Decoding

As shown in Figure 2, we adopt a block-wise decoding strategy to support the use of a Key-Value (KV) Cache. Initially, we compute and store the KV Cache for the prompt, which is reused throughout Block 0. Within each block, the same cache is reused for multiple decoding steps. After completing the decoding of a block, we update the cache for all tokens (not just the newly generated ones). This cache update can be performed jointly with the decoding step, so compared to not using caching, there is no additional computational overhead. This approach results in an approximate decoding process, due to the use of full attention in masked diffusion models (Nie et al., 2025b; Ye et al., 2025).

The effectiveness of our approximate KV Cache approach stems from the observation that KV activations exhibit high similarity across adjacent inference steps, as illustrated in Figure 3. The red boxed region in Figure 3a highlights the similarity scores within a block, which are consistently close to 1. This indicates that the differences in prefix keys and values during block decoding are negligible, allowing us to safely reuse the cache without significant loss in accuracy.

## 3.3 Confidence-Aware Parallel Decoding

Furthermore, we implement a bidirectional version of our KV caching mechanism, named DualCache, that caches not only the prefix tokens but also the suffix tokens, which consist entirely of masked tokens under our block-wise decoding scheme. As shown in Table 4, DualCache results in further acceleration. The red boxed region in Figure 3b further demonstrates that the differences in suffix keys and values during block decoding are negligible.

While approaches like employing auxiliary models to explicitly capture these dependencies exist (Liu et al., 2024; Xu et al., 2024), they typically increase the complexity of the overall pipeline. In contrast to these approaches, we propose a simple-yet-effective confidence-aware decoding algorithm designed to mitigate this conditional independence issue.

Concretely, at each iteration, rather than aggressively unmasking all masked tokens using their independent marginal probabilities, we compute a confidence score for each token (e.g., the maximum softmax probability). Only those with confidence exceeding a threshold are unmasked in the current step; the rest remain masked and are reconsidered in future steps. If no token's confidence exceeds the threshold, we always unmask the token with the highest confidence to ensure progress and prevent an infinite loop. This strategy accelerates generation while reducing errors from uncertain or ambiguous predictions.

A critical question, however, is: *When is it theoretically justifiable to decode tokens in parallel using independent marginals, despite the true joint distribution potentially containing dependencies?* We address this with the following formal result, which characterizes the conditions under which greedy parallel (product of marginal distribution) decoding is equivalent to greedy sequential (true joint distribution) decoding in the high-confidence regime, and quantifies the divergence between the two distributions.

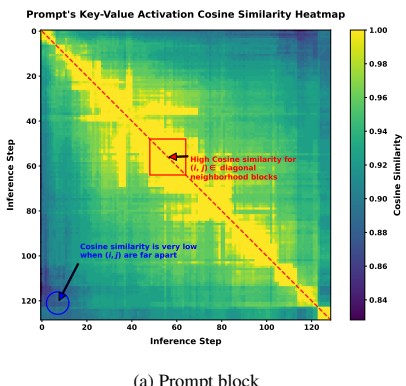 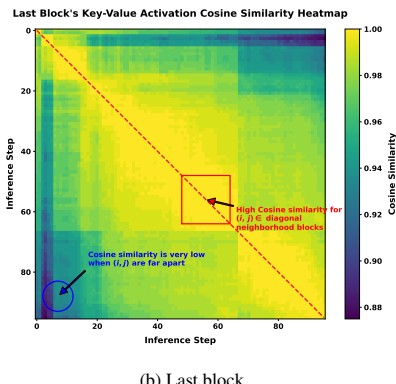

(a) Prompt block                                    (b) Last block

Figure 3: **Heatmaps of Key-Value Activation Cosine Similarity Across Inference Steps in LLaDA-Instruct.** Cosine similarity heatmaps of Key-Value activations for (a) the prompt and (b) the last (suffix) block. High similarity along the diagonal ($i \approx j$, red boxes) indicates that activations for adjacent inference steps are highly similar. This supports using an approximate block-wise KV Cache, allowing cached activations to be reused for faster decoding with negligible impact on accuracy.

Prior to presenting the theorem, we will define the mathematical notation used in its statement. Let $p_{\boldsymbol{\theta}}(\cdot|E)$ denote the conditional probability mass function (PMF) given by an MDM condition on $E$ (comprising a prompt $p_0$ and previously generated tokens). Suppose the model is to predict $n$ tokens for positions $i_1, \ldots, i_n$ not in $E$. Let $\boldsymbol{X} = (X_{i_1}, \ldots, X_{i_n})$ be the vector of $n$ tokens, where each $X_{i_j}$ takes values in vocabulary $\mathcal{V}$. Let $p(\boldsymbol{X}|E) \equiv p_{\boldsymbol{\theta}}(X_{i_1}, \ldots, X_{i_n}|E)$ be the joint conditional PMF according to the model. Let $p_j(X_{i_j}|E) \equiv p_{\boldsymbol{\theta}}(X_{i_j}|E)$ be the marginal conditional PMF for position $i_j$. Parallel decoding generates tokens using the product of marginals: $q(\boldsymbol{X}|E) = \prod_{j=1}^{n} p_j(X_{i_j}|E)$. The proof of Theorem 1 and relevant discussions are in Appendix A.

**Theorem 1** (Parallel Decoding under High Confidence). *Suppose there exists a specific sequence of tokens $\boldsymbol{x}^* = (x_{i_1}, \ldots, x_{i_n})$ such that for each $j \in \{1, \ldots, n\}$, the model has high confidence in $x_{i_j}$: $p_j(X_{i_j} = x_{i_j}|E) > 1 - \epsilon$ for some small $\epsilon > 0$. Then, the following results hold:*

*1. Equivalence for Greedy Decoding: If $(n+1)\epsilon \leq 1$ (i.e., $\epsilon \leq \frac{1}{n+1}$), then*

$$\underset{\boldsymbol{z}}{\arg\max}\, p(\boldsymbol{z}|E) = \underset{\boldsymbol{z}}{\arg\max}\, q(\boldsymbol{z}|E) = \boldsymbol{x}^*. \tag{4}$$

*This means that greedy parallel decoding (selecting $\arg\max q$) yields the same result as greedy sequential decoding (selecting $\arg\max p$).*

*This bound is tight: if $\epsilon > \frac{1}{n+1}$, there exist distributions $p(\boldsymbol{X}|E)$ satisfying the high-confidence marginal assumption for which $\arg\max_{\boldsymbol{z}} p(\boldsymbol{z}|E) \neq \arg\max_{\boldsymbol{z}} q(\boldsymbol{z}|E)$.*

*2. Distance and Divergence Bounds: Let $p(\cdot|E)$ and $q(\cdot|E)$ be denoted as $p$ and $q$ for brevity.*

*$L_p$ Distance ($p \geq 1$): For $n > 1$, $D_p(p, q) < ((n-1)^p + 2n)^{1/p}\epsilon$. Specifically, for Total Variation Distance ($D_{TV}(p, q) = \frac{1}{2}D_1(p, q)$): $D_{TV}(p, q) < \frac{3n-1}{2}\epsilon$.*

*Forward KL Divergence: For $n > 1$, $D_{\mathrm{KL}}(p\|q) < (n-1)(H_b(\epsilon) + \epsilon \ln(|\mathcal{V}| - 1))$, where $H_b(\epsilon) = -\epsilon \ln \epsilon - (1-\epsilon)\ln(1-\epsilon)$ is the binary entropy function, and $|\mathcal{V}|$ is the size of the vocabulary.*

Building on this theorem, we propose a practical *factor*-based parallel decoding strategy as an extension of the threshold strategy that adaptively selects how many tokens to decode in parallel based on the confidence levels. Concretely, given the model's marginal confidence estimates for $n$ tokens in a block, we sort these confidences and select the largest $n$ such that $(n+1)(1 - c^{(n)}) < f$, where $f$ is a fixed decoding factor hyperparameter and $c^{(n)}$ is the $n$-th highest confidence. At each step, the top-$n$ tokens are decoded in parallel. This formulation mirrors the bound in Theorem 1 and ensures that decoding only proceeds when the marginal confidence is sufficiently high to approximate the joint decoding reliably. In contrast to the static threshold-based strategy, factor-based decoding dynamically controls the degree of parallelism in a theoretically grounded manner.

---

**Algorithm 1** Block-wise Confidence-aware Parallel Decoding with (Dual) KV Cache

---

**Require:** $p_{\boldsymbol{\theta}}$, prompt $p_0$, answer length $L$, blocks $K$, block size $B$, steps per block $T$, threshold $\tau$,
  use_DualCache, strategy $\in \{$threshold, factor$\}$, factor $f$
1: $x \leftarrow [p_0; \text{[MASK]}, ..., \text{[MASK]}]$
2: **Initialize KV Cache** (single or dual) for $x$ (fuse with decoding).         *// KV Cache Init*
3: **for** $k = 1$ to $K$ **do**
4:    $s \leftarrow |p_0| + (k-1)B, \ e \leftarrow |p_0| + kB$
5:    **for** $t = 1$ to $T$ **do**
6:       Use cache, run $p_{\boldsymbol{\theta}}$ on $x^{[s,e]}$ if use_DualCache else $x^{[s,:]}$     *// Cache Reuse*
7:       For masked $x^i$, compute confidence $c^i = \max_x p_{\boldsymbol{\theta}}(x^i|\cdot)$     *// Confidence scoring*
8:       **if** strategy == threshold **then**
9:          Unmask all $i$ in $[s,e]$ with $c^i \geq \tau$, always unmask max $c^i$
10:       **else if** strategy == factor **then**
11:          Sort $c^i$ in descending order as $(c^{(1)}, c^{(2)}, ..., c^{(m)})$
12:          Find largest $n$ such that $(n+1)(1 - c^{(n)}) < f$
13:          Unmask top-$n$ tokens, always unmask the max $c^i$
14:       **end if**
15:       **if** all $x^{[s,e]}$ unmasked **then**
16:          **break**
17:       **end if**
18:    **end for**
19:    **Update KV cache**: **if** use_DualCache**:** prefix & suffix; **else:** prefix.    *// Cache Update*
20: **end for**
21: **return** $x$

---

Table 1: Comprehensive benchmark results on the LLaDA-Instruct suite. Each cell presents the accuracy and the decoding throughput in tokens per second with relative speedup to the LLaDA baseline (bottom row, blue: tokens per second/orange: relative speedup). The highest throughput and speedup for each configuration are highlighted.

| Benchmark | Gen Length | LLaDA | +Cache | +Parallel | +Cache+Parallel (Fast-dLLM) |
|---|---|---|---|---|---|
| GSM8K (5-shot) | 256 | 79.3 | 79.5 | 79.2 | 78.5 |
| | | 6.7 (1×) | 21.2 (3.2×) | 16.5 (2.5×) | 54.4 (8.1×) |
| | 512 | 77.5 | 77.0 | 77.6 | 77.2 |
| | | 3.2 (1×) | 10.4 (3.3×) | 18.6 (5.8×) | 35.3 (11.0×) |
| MATH (4-shot) | 256 | 33.5 | 33.3 | 33.4 | 33.2 |
| | | 9.1 (1×) | 23.7 (2.6×) | 24.8 (2.7×) | 51.7 (5.7×) |
| | 512 | 37.2 | 36.2 | 36.8 | 36.0 |
| | | 8.0 (1×) | 19.7 (2.5×) | 23.8 (3.0×) | 47.1 (5.9×) |
| HumanEval (0-shot) | 256 | 41.5 | 42.7 | 43.9 | 43.3 |
| | | 30.5 (1×) | 40.7 (1.3×) | 101.5 (3.3×) | 114.1 (3.7×) |
| | 512 | 43.9 | 45.7 | 43.3 | 44.5 |
| | | 18.4 (1×) | 29.3 (1.6×) | 57.1 (3.1×) | 73.7 (4.0×) |
| MBPP (3-shot) | 256 | 29.4 | 29.6 | 28.4 | 28.2 |
| | | 6.0 (1×) | 17.0 (2.8×) | 24.8 (4.1×) | 44.8 (7.5×) |
| | 512 | 14.8 | 13.4 | 15.0 | 13.8 |
| | | 4.3 (1×) | 10.1 (2.3×) | 22.3 (5.1×) | 39.5 (9.2×) |
| IFEval | 256 | 57.1 | 53.6 | 57.1 | 54.0 |
| | | 15.7 (1×) | 19.7 (1.3×) | 30.9 (2.0×) | 36.0 (2.3×) |
| | 512 | 58.0 | 57.5 | 57.9 | 57.3 |
| | | 8.2 (1×) | 12.0 (1.5×) | 22.2 (2.7×) | 30.0 (3.7×) |

## 4 EXPERIMENTS

### 4.1 EXPERIMENTAL SETUP

All experiments are conducted on an NVIDIA A100 80GB GPU. The proposed approach, Fast-dLLM, comprises two components: a Key-Value Cache mechanism and a Confidence-Aware Parallel Decoding strategy. The KV Cache component introduces a hyperparameter, the cache block size, varied between 4 and 32. The parallel decoding strategy uses a confidence threshold hyperparameter, explored in the range of 0.5 to 1.0. Unless otherwise specified, we use PrefixCache with block size of 32 and the threshold to 0.9.

Table 2: Comprehensive benchmark results on Dream-Base variants over four tasks with different generation lengths (256 and 512). Each cell shows accuracy (top row) and decoding throughput in tokens per second with relative speedup to Dream-Base baseline (bottom row, blue: tokens per second/orange: relative speedup). Numbers in yellow indicate the highest throughput and speedup per configuration.

| Benchmark | Gen Length | Dream | +Cache | +Parallel | +Cache+Parallel (Fast-dLLM) |
|---|---|---|---|---|---|
| GSM8K (5-shot) | 256 | 75.0 | 74.3 | 74.2 | 74.8 |
| | | 9.1 (1×) | 32.5 (3.6×) | 14.2 (1.6×) | 48.2 (5.3×) |
| | 512 | 76.0 | 74.3 | 73.4 | 74.0 |
| | | 7.7 (1×) | 25.6 (3.3×) | 14.6 (1.9×) | 42.9 (5.6×) |
| MATH (4-shot) | 256 | 38.4 | 36.8 | 37.9 | 37.6 |
| | | 11.4 (1×) | 34.3 (3.0×) | 27.3 (2.4×) | 66.8 (5.9×) |
| | 512 | 39.8 | 38.0 | 39.5 | 39.3 |
| | | 9.6 (1×) | 26.8 (2.8×) | 31.6 (3.2×) | 63.3 (6.5×) |
| HumanEval (0-shot) | 256 | 49.4 | 53.7 | 49.4 | 54.3 |
| | | 23.3 (1×) | 35.2 (1.5×) | 45.6 (2.0×) | 62.0 (2.8×) |
| | 512 | 54.3 | 54.9 | 51.8 | 54.3 |
| | | 16.3 (1×) | 27.8 (1.7×) | 29.8 (1.8×) | 52.8 (3.2×) |
| MBPP (3-shot) | 256 | 56.6 | 53.2 | 53.8 | 56.4 |
| | | 11.2 (1×) | 34.5 (3.1×) | 31.8 (2.8×) | 76.0 (6.8×) |
| | 512 | 55.6 | 53.8 | 55.4 | 55.2 |
| | | 9.4 (1×) | 26.7 (2.8×) | 37.6 (4.0×) | 73.6 (7.8×) |

We evaluate Fast-dLLM on two recent diffusion-based language models: LLaDA (Nie et al., 2025b), LLaDA-1.5 (Zhu et al., 2025) and Dream (Ye et al., 2025). Benchmarks include four widely-used datasets—GSM8K, MATH, HumanEval, MBPP and IFEval, to assess performance across diverse reasoning and code generation tasks. We also test under varying generation lengths to evaluate scalability and robustness.

In addition, we extend our evaluation to LLaDA-V (You et al., 2025), a multimodal variant of LLaDA tailored for vision-language reasoning tasks. For this, we use two challenging multimodal benchmarks: MathVista and MathVerse, which require solving math problems grounded in complex visual scenes.

Inference throughput is measured as the average number of output tokens generated per second, calculated over the full sequence until the end-of-sequence (<eos>) token is reached. This metric reflects true end-to-end decoding speed. All evaluations are conducted using the standardized lm-eval library to ensure consistency and reproducibility.

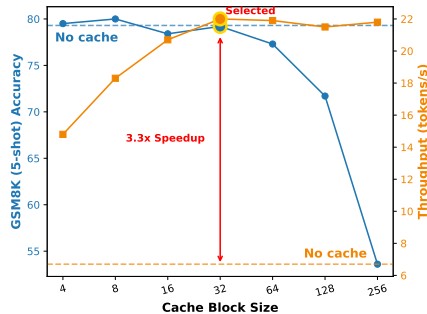

Figure 4: **Impact of Cache Block Size on Accuracy and Throughput.** The orange line illustrates the effect of varying cache block size on throughput, while the blue line depicts accuracy.

## 4.2 MAIN RESULTS: PERFORMANCE AND SPEED

We evaluate Fast-dLLM on LLaDA-Instruct and Dream-Base across four benchmarks (Tables 1 and 2).

Introducing the KV Cache yields 2×–3.6× speedup over the vanilla backbone across tasks and lengths. Parallel decoding alone provides further acceleration, typically 4×–6×, especially at longer generation lengths.

Combining both gives the largest gains. On LLaDA, throughput improves up to 11× (GSM8K, len 512) and 9.2× (MBPP, len 512). On Dream-Base, maximum gains are 7.8× (MBPP, len 512) and 5.6× (GSM8K, len 512). These results show the methods are not only effective individually but also highly complementary.

Importantly, these efficiency gains come with negligible accuracy cost: across all benchmarks accuracy remains within 1–2 points of the backbone, and in some cases even improves. Longer sequences, common in few-shot and code generation, benefit disproportionately due to greater cache

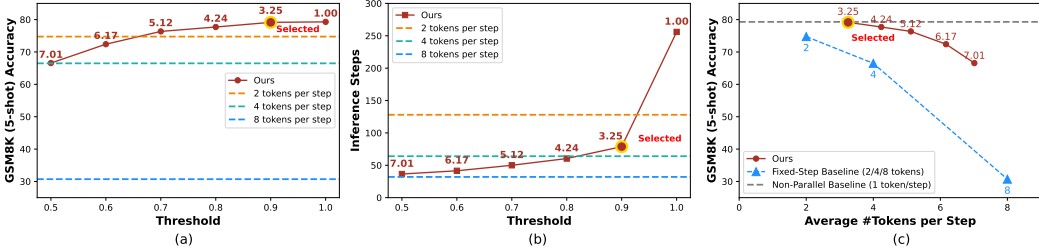

Figure 5: (a) The red line shows the GSM8K (5-shot) accuracy across different confidence thresholds. Numbers along the red line indicate the average number of tokens decoded at each step. The three dashed lines represent the accuracy of the baseline method when selecting the top 2, 4, or 8 tokens per step. (b) The number of inference steps required under varying confidence thresholds. (c) A comparison between our method and the baseline on GSM8K (5-shot) accuracy, plotted against the average number of tokens per step. Our method consistently outperforms the baseline.

Table 3: **Performance and Speedup Comparison of LLaDA-V on MathVista and MathVerse.** Each benchmark includes results from Full Steps, Half Steps, and Fast-dLLM. Fast-dLLM significantly improves throughput (highlighted), with minimal accuracy loss.

| Metric | MathVista | | | MathVerse | | |
|---|---|---|---|---|---|---|
| | Full Steps | Half Steps | Fast-dLLM | Full Steps | Half Steps | Fast-dLLM |
| Accuracy (%) | 59.2 | 59.7 | 56.6 | 28.5 | 28.3 | 28.6 |
| Throughput (Speedup) | 2.84 (1×) | 5.56 (1.96×) | 28.2 (9.9×) | 2.75 (1×) | 5.17 (1.88×) | 23.3 (8.5×) |

reuse and batch computation. We also evaluate LLaDA-1.5, which achieves consistently higher accuracy and comparable or better throughput (Table 12).

Beyond text-only models, we test Fast-dLLM on multimodal LLaDA-V using MathVista and Math-Verse. LLaDA-V is sensitive to block size, losing over 8% accuracy when reduced from 96 to 8 on MathVista. To mitigate this, we retain full block length and apply refresh-based updates, yielding up to $9.9\times$ speedup with minimal degradation (Table 3). On MathVerse, Fast-dLLM even slightly improves accuracy, showing robustness on vision-language reasoning.

Overall, improvements hold across architectures (LLaDA, Dream), task types (math reasoning, program synthesis), and modalities (text, vision), establishing Fast-dLLM as a broadly applicable framework for accelerating masked diffusion LLMs.

## 4.3 ABLATIONS AND ANALYSIS

Table 4: **Performance and Speedup Comparison on LLaDA Between 5-Shot and 8-Shot Settings at Generation Length 1024.** This table compares the accuracy and throughput speedups of different decoding strategies under 5-shot and 8-shot configurations using a generation length of 1024. The results demonstrate how increased prefill length enhances the effectiveness of caching strategies, particularly for DualCache.

Table 5: **Impact of Generation Length on Accuracy and Speedup Under 8-Shot for LLaDA.** This table illustrates the effect of varying generation lengths (256, 512, and 1024) on decoding performance and efficiency for different caching strategies under the 8-shot setting. Longer generation lengths lead to higher throughput gains, especially for DualCache, validating the scalability of our approach.

| Setting. | LLaDA | Parallel Decoding | | |
|---|---|---|---|---|
| | | No Cache | PrefixCache | DualCache |
| 5-shot | 77.0 | 77.4 | 75.2 | 74.7 |
| | 1.1 (1×) | 11.7 (10.6×) | 14.4 (13.1×) | 21.6 (19.6×) |
| 8-shot | 77.3 | 78.0 | 75.7 | 76.0 |
| | 0.7 (1×) | 9.3 (13.3×) | 13.0 (18.6×) | 19.3 (27.6×) |

| Len. | LLaDA | Parallel Decoding | | |
|---|---|---|---|---|
| | | No Cache | PrefixCache | DualCache |
| 256 | 77.6 | 77.9 | 77.3 | 76.9 |
| | 4.9 (1×) | 16.4 (3.3×) | 49.2 (10.0×) | 46.3 (9.4×) |
| 512 | 78.9 | 78.9 | 74.8 | 75.4 |
| | 2.3 (1×) | 14.0 (6.1×) | 32.0 (13.9×) | 36.4 (15.8×) |
| 1024 | 77.3 | 78.0 | 75.7 | 76.0 |
| | 0.7 (1×) | 9.3 (13.3×) | 13.0 (18.6×) | 19.3 (27.6×) |

We perform extensive ablations to assess the contribution of different components in Fast-dLLM, focusing on prefill length, generation length, cache variants, block size, and confidence thresholds.

**Prefill and Generation Length** Tables 4 and 5 show that longer prefill (*n*-shot) and generation lengths significantly boost speedup. For example, DualCache improves from $19.6\times$ (5-shot, gen len 1024) to $27.6\times$ (8-shot, gen len 1024). Speedup grows as generation length increases (e.g., $9.4\times$ at 256 tokens vs. $27.6\times$ at 1024), consistent with amortizing computation over longer sequences.

**Prefix KV Cache vs. DualCache** DualCache generally surpasses prefix KV Cache, especially for long generations (e.g., $27.6\times$ vs. $18.6\times$ at gen len 1024, Table 5). Accuracy remains competitive, confirming DualCache's ability to exploit parallelism and cache locality effectively.

**Cache Block Size** Figure 4 shows that smaller block sizes maximize accuracy but add cache-update overhead, while larger sizes risk mismatch. Block size 32 achieves the best trade-off, balancing throughput and accuracy.

**Dynamic Threshold vs. Fixed Token-per-Step** On GSM8K (Figure 5), our confidence-aware strategy consistently outperforms fixed baselines: it yields higher accuracy with comparable or fewer NFEs, generates more tokens per step, and approaches the accuracy of the 1-token baseline with much higher throughput.

**Factor Decoding vs. Fixed Strategies** As shown in Figure 8 and Table 11, factor-based decoding achieves competitive or better accuracy with fewer steps. Larger factors decode more tokens per step, reducing iterations while preserving performance. Compared to threshold decoding, factor decoding maintains accuracy but achieves higher throughput via adaptive granularity (see Appendix C.4).

**Decoding Efficiency and Limitations** Section C.5 shows PrefixCache accelerates diffusion-based LLMs like LLaDA by up to $5\times$ in compute-bound settings, reaching or exceeding LLaMA throughput at small batch sizes. However, at larger batches it falls behind LLaMA, since diffusion models incur higher overhead from full attention during decoding.

## 5 RELATED WORK

We put a short version here, the full **Related Work** is in Appendix D.
**Diffusion LLM.** Diffusion models, proven first in vision/audio, are re-shaping text generation. Discrete formulations re-cast noising/denoising as Markov chains (Austin et al., 2021), continuous-time processes (Campbell et al., 2022), score-matching (Lou et al., 2023) or masked-language tasks (Shi et al., 2024; Sahoo et al., 2024; Zheng et al., 2024); all maximize ELBO or entropy objectives and can be made equivalent. Plugging the masked objective into BART or LLaMA yields Diffusion-NAT (Zhou et al., 2023), LLaDA / DiffuLLaMA (Nie et al., 2025b; Gong et al., 2024) and Dream (Ye et al., 2025): 7 B-parameter diffusion LLMs that refine corrupted sequences in parallel and match AR quality while promising >10× speed-up.
**LLM Acceleration.** KV-cache (Vaswani, 2017) avoids recomputation in AR Transformers, but full-sequence diffusion invalidates it; Block-diffusion (Arriola et al., 2025) restores the cache by generating block-by-block. **Non-autoregressive (NAR)** decoding outputs many tokens at once (Xiao et al., 2023), yielding large latency gains yet lower quality. Diffusion LLMs are a new NAR family, but until now the speed benefit has been offset by accuracy loss (Nie et al., 2025b).

## 6 CONCLUSION

We address key inefficiencies in Diffusion-based Large Language Models (Diffusion LLMs), which traditionally lack KV Cache support and suffer from degraded performance in parallel decoding. To bridge the gap with autoregressive models, we propose Fast-dLLM, a framework introducing an approximate KV Cache tailored to bidirectional attention via block-wise generation. We further mitigate token-dependency issues in parallel decoding with a Confidence-Aware strategy that enables safe, efficient multi-token generation. Experiments across benchmarks and baselines (LLaDA, Dream) demonstrate up to $27.6\times$ speedup with minimal accuracy loss, establishing Fast-dLLM as a practical path toward making Diffusion LLMs competitive for real-world deployment.

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

# A PROOF

In this section, we will give the comprehensive proof and discussion of Theorem 1.

*Proof.* **Step 1: Show that $x^*$ is the unique maximizer of $q(x)$.**

Let $p_j^* = p_j(X_{i_j} = x_{i_j}|E)$. We are given $p_j^* > 1 - \epsilon$. Let $\epsilon'_j = 1 - p_j^* = p_j(X_{i_j} \neq x_{i_j}|E)$. Thus, $\epsilon'_j < \epsilon$. The product-of-marginals probability mass function (PMF) is

$$q(\boldsymbol{z}|E) = \prod_{j=1}^{n} p_j(X_{i_j} = z_j|E).$$

To maximize $q(\boldsymbol{z}|E)$, we must maximize each term $p_j(X_{i_j} = z_j|E)$ independently. The condition $(n+1)\epsilon \leq 1$ implies $\epsilon \leq 1/(n+1)$. Since $n \geq 1$, it follows that $1/(n+1) \leq 1/2$. So, $\epsilon \leq 1/2$. Therefore, for the chosen $x_{i_j}$:

$$p_j^* = p_j(X_{i_j} = x_{i_j}|E) > 1 - \epsilon \geq 1 - 1/2 = 1/2.$$

This means $x_{i_j}$ is the unique maximizer for $p_j(\cdot|E)$. So,

$$\operatorname*{argmax}_{\boldsymbol{z}} q(\boldsymbol{z}|E) = (x_{i_1}, \ldots, x_{i_n}) = \boldsymbol{x}^*.$$

**Step 2: Show that $x^*$ is the unique maximizer of $p(x)$.**

We want to show $p(\boldsymbol{x}^*|E) > p(\boldsymbol{z}|E)$ for all $\boldsymbol{z} \neq \boldsymbol{x}^*$. Using the Bonferroni inequality:

$$p(\boldsymbol{x}^*|E) = p(\cap_{j=1}^{n}\{X_{i_j} = x_{i_j}\}|E) \geq 1 - \sum_{j=1}^{n} p(X_{i_j} \neq x_{i_j}|E) = 1 - \sum_{j=1}^{n} \epsilon'_j.$$

Since $\epsilon'_j < \epsilon$ for all $j$, we have $\sum_{j=1}^{n} \epsilon'_j < n\epsilon$. So,

$$p(\boldsymbol{x}^*|E) > 1 - n\epsilon.$$

Now consider any $\boldsymbol{z} = (z_1, \ldots, z_n)$ such that $\boldsymbol{z} \neq \boldsymbol{x}^*$. This means there is at least one index $k$ such that $z_k \neq x_{i_k}$. The event $\{\boldsymbol{X} = \boldsymbol{z}\}$ is a sub-event of $\{X_{i_k} = z_k\}$. So,

$$p(\boldsymbol{z}|E) \leq p_k(X_{i_k} = z_k|E).$$

Since $z_k \neq x_{i_k}$,

$$p_k(X_{i_k} = z_k|E) \leq p_k(X_{i_k} \neq x_{i_k}|E) = \epsilon'_k < \epsilon.$$

Thus,

$$p(\boldsymbol{z}|E) < \epsilon.$$

For $p(\boldsymbol{x}^*|E) > p(\boldsymbol{z}|E)$ to hold, it is sufficient that

$$1 - n\epsilon \geq \epsilon,$$

which simplifies to $1 \geq (n+1)\epsilon$, or $\epsilon \leq \frac{1}{n+1}$. The theorem assumes $(n+1)\epsilon < 1$, which is exactly this condition. The strict inequalities $p(\boldsymbol{x}^*|E) \geq 1 - \sum \epsilon'_j > 1 - n\epsilon$ and $p(\boldsymbol{z}|E) \leq \epsilon'_k < \epsilon$ ensure that $p(\boldsymbol{x}^*|E) > p(\boldsymbol{z}|E)$. Thus,

$$\operatorname*{argmax}_{\boldsymbol{z}} p(\boldsymbol{z}|E) = \boldsymbol{x}^*.$$

Combined with the argmax of $q$, this proves the main statement of Part 1:

$$\operatorname*{argmax}_{\boldsymbol{z}} p(\boldsymbol{z}|E) = \operatorname*{argmax}_{\boldsymbol{z}} q(\boldsymbol{z}|E) = \boldsymbol{x}^*.$$

**Step 3: Tightness of the bound $\frac{1}{n+1}$.**

The bound $\epsilon \leq \frac{1}{n+1}$ is tight. This means if $\epsilon > \frac{1}{n+1}$, one can construct a scenario where the marginal conditions $p_j(X_{i_j} = x_{i_j}|E) > 1 - \epsilon$ hold, but $\operatorname{argmax}_{\boldsymbol{z}} p(\boldsymbol{z}|E) \neq \boldsymbol{x}^*$ (which is $\operatorname{argmax}_{\boldsymbol{z}} q(\boldsymbol{z}|E)$ as long as $\epsilon \leq 1/2$).

Consider a vocabulary $\mathcal{V} = \{0, 1\}$ and let $x_{i_j} = 0$ for all $j$, so $\boldsymbol{x}^* = (0, \ldots, 0)$. For each $j \in \{1, \ldots, n\}$, let $\mathbf{e}_j$ be the vector with 1 at position $j$ and 0 elsewhere. Let $\eta = \frac{1}{n+1}(\epsilon - \frac{1}{n+1}) > 0$. Set $p(\mathbf{e}_j|E) = \frac{1}{n+1} + \frac{1}{n}\eta$, $\forall 1 \le j \le n$ and $p(\boldsymbol{x}^*|E) = \frac{1}{n+1} - \eta$, then $\boldsymbol{x}^* \notin \operatorname{argmax}_{\boldsymbol{z}} p(\boldsymbol{z}|E)$. The marginal probabilities are:

$$p_j(X_{i_j} = 1|E) = p(\mathbf{e}_j|E) = \frac{1}{n+1} + \frac{1}{n}\eta, \ \forall 1 \le j \le n.$$

$$p_j(X_{i_j} = 0|E) = 1 - p_j(X_{i_j} = 1|E) = 1 - \epsilon_c = \frac{n}{n+1} - \frac{1}{n}\eta > 1 - \epsilon,$$

because

$$\frac{1}{n}\eta = \frac{1}{n(n+1)}(\epsilon - \frac{1}{n+1}) < \epsilon - \frac{1}{n+1}$$

So, the marginal condition $p_j(X_{i_j} = x_{i_j}|E) > 1 - \epsilon$ (with $x_{i_j} = 0$) holds. As shown, $\operatorname{argmax}_{\boldsymbol{z}} p(\boldsymbol{z}|E)$ can be made different from $\boldsymbol{x}^*$. Thus, if $\epsilon > \frac{1}{n+1}$, the argmax of $p$ and $q$ may not be the same.

**Step 4: Bound the $L_p$ distance.** Let $A_j$ be the event $\{X_{i_j} = x_{i_j}\}$.

$$D_p(p, q)^p = |p(\boldsymbol{x}^*|E) - q(\boldsymbol{x}^*|E)|^p + \sum_{\boldsymbol{z} \neq \boldsymbol{x}^*} |p(\boldsymbol{z}|E) - q(\boldsymbol{z}|E)|^p.$$

The term $|p(\cap_{j=1}^n A_j|E) - \prod_{j=1}^n p(A_j|E)|$ (using $p(A_j|E)$ for $p_j(X_{i_j} = x_{i_j}|E)$) can be bounded. Since

$$1 - \sum_{j=1}^n \epsilon'_j \le p(\cap_{j=1}^n A_j|E) \le \min_{1 \le j \le n} p(A_j|E) = 1 - \max_{1 \le j \le n} \epsilon'_j,$$

$$1 - \sum_{j=1}^n \epsilon'_j \le \prod_{j=1}^n (1 - \epsilon'_j) = \prod_{j=1}^n p(A_j|E) \le 1 - \max_{1 \le j \le n} \epsilon'_j.$$

Thus,

$$|p(\boldsymbol{x}^*|E) - q(\boldsymbol{x}^*|E)| < (n-1)\epsilon.$$

For $\boldsymbol{z} \neq \boldsymbol{x}^*$: $p(\boldsymbol{z}|E) < \epsilon$ and $q(\boldsymbol{z}|E) < \epsilon$. So,

$$|p(\boldsymbol{z}|E) - q(\boldsymbol{z}|E)| < \epsilon.$$

The sum $\sum_{\boldsymbol{z} \neq \boldsymbol{x}^*} |p(\boldsymbol{z}|E) - q(\boldsymbol{z}|E)|$ can be bounded:

$$\sum_{\boldsymbol{z} \neq \boldsymbol{x}^*} |p(\boldsymbol{z}|E) - q(\boldsymbol{z}|E)| \le \sum_{\boldsymbol{z} \neq \boldsymbol{x}^*} (p(\boldsymbol{z}|E) + q(\boldsymbol{z}|E)) = p(\boldsymbol{X} \neq \boldsymbol{x}^*|E) + q(\boldsymbol{X} \neq \boldsymbol{x}^*|E).$$

$$p(\boldsymbol{X} \neq \boldsymbol{x}^*|E) = 1 - p(\boldsymbol{x}^*|E) < 1 - (1 - \sum_{j=1}^n \epsilon'_j) = \sum_{j=1}^n \epsilon'_j < n\epsilon.$$

$$q(\boldsymbol{X} \neq \boldsymbol{x}^*|E) = 1 - q(\boldsymbol{x}^*|E) < 1 - \prod_{j=1}^n (1 - \epsilon'_j) \le \sum_{j=1}^n \epsilon'_j < n\epsilon.$$

So,

$$\sum_{\boldsymbol{z} \neq \boldsymbol{x}^*} |p(\boldsymbol{z}|E) - q(\boldsymbol{z}|E)| < 2n\epsilon.$$

Then,

$$\sum_{\boldsymbol{z} \neq \boldsymbol{x}^*} |p(\boldsymbol{z}|E) - q(\boldsymbol{z}|E)|^p \le (\sup_{\boldsymbol{z} \neq \boldsymbol{x}^*} |p(\boldsymbol{z}|E) - q(\boldsymbol{z}|E)|)^{p-1} \sum_{\boldsymbol{z} \neq \boldsymbol{x}^*} |p(\boldsymbol{z}|E) - q(\boldsymbol{z}|E)|$$

$$< \epsilon^{p-1}(2n\epsilon) = 2n\epsilon^p.$$

Therefore,

$$D_p(p, q)^p < ((n-1)\epsilon)^p + 2n\epsilon^p = ((n-1)^p + 2n)\epsilon^p.$$

So,
$$D_p(p,q) < ((n-1)^p + 2n)^{1/p}\epsilon.$$
For $p = 1$,
$$D_1(p,q) < (n-1+2n)\epsilon = (3n-1)\epsilon.$$
And for Total Variation Distance,
$$D_{TV}(p,q) = \frac{1}{2}D_1(p,q) < \frac{3n-1}{2}\epsilon.$$

**Step 4: Bound the forward KL divergence.**

$$D_{\mathrm{KL}}(p\|q) = \sum_{\boldsymbol{z}} p(\boldsymbol{z}|E)\log\frac{p(\boldsymbol{z}|E)}{q(\boldsymbol{z}|E)} = I(X_{i_1};\ldots;X_{i_n}|E).$$
The conditional total correlation can be expanded using the chain rule:
$$I(X_{i_1};\ldots;X_{i_n}|E) = \sum_{k=2}^{n} I(X_{i_k};X_{i_1},\ldots,X_{i_{k-1}}|E).$$
Each term is bounded by the conditional entropy:
$$I(X_{i_k};X_{i_1},\ldots,X_{i_{k-1}}|E) \le H(X_{i_k}|E).$$
The conditional entropy $H(X_{i_k}|E)$ is bounded. Since $p_k(X_{i_k} = x_{i_k}|E) > 1 - \epsilon$, it implies $p_k(X_{i_k} \ne x_{i_k}|E) = \epsilon'_k < \epsilon$. The entropy is maximized when the remaining probability $\epsilon'_k$ is spread uniformly, leading to:
$$H(X_{i_k}|E) \le H_b(\epsilon'_k) + \epsilon'_k \ln(|\mathcal{V}| - 1) < H_b(\epsilon) + \epsilon \ln(|\mathcal{V}| - 1).$$
Summing $(n-1)$ such terms (for $k = 2, \ldots, n$):
$$D_{\mathrm{KL}}(p\|q) < (n-1)[H_b(\epsilon) + \epsilon \ln(|\mathcal{V}| - 1)].$$

$\square$

**Remark 1.** *Assumption of a Well-Defined Joint $p_{\boldsymbol{\theta}}(X_{i_1}, \ldots, X_{i_n}|E)$: The theorem and proof rely on $p_{\boldsymbol{\theta}}(X_{i_1}, \ldots, X_{i_n}|E)$ being a well-defined joint probability mass function from which the marginals $p_{\boldsymbol{\theta}}(X_{i_j}|E)$ are consistently derived. This implies that the joint PMF is coherent and its definition does not depend on a specific factorization order beyond what is captured by the conditioning on $E$. In practice, while MDM may not strictly satisfy this property, its behavior typically offers a close approximation. The theorem holds for an idealized $p_{\boldsymbol{\theta}}$ that possesses these properties. As MDMs become larger and more powerful, their learned distributions might better approximate such consistency.*

***Worst-Case Analysis:*** *The conditions and bounds provided in the theorem (e.g., $(n+1)\epsilon \le 1$) are derived from a worst-case analysis. This means the bounds are guaranteed to hold if the conditions are met, regardless of the specific structure of $p_{\boldsymbol{\theta}}(\boldsymbol{X}|E)$ beyond the high-confidence marginal property. In practice, the actual case might be "better behaved" than the worst-case scenario. For instance, the dependencies between $X_{i_j}$ and $X_{i_k}$ (given $E$) might be weaker than what the worst-case construction assumes. Consequently, the argmax equivalence (Result 1) might still hold frequently even if $(n+1)\epsilon$ is slightly greater than 1 (but not much larger). The condition identifies a threshold beyond which guarantees break down in the worst case, but practical performance can be more robust. Similarly, the actual $L_p$ distances or KL divergence might be smaller than the upper bounds suggest if the true joint $p_{\boldsymbol{\theta}}(\boldsymbol{X}|E)$ is closer to the product of marginals $q(\boldsymbol{X}|E)$ than the worst-case configurations.*

## B CASE STUDY

### B.1 EFFECT OF CACHING STRATEGIES ON RESPONSE QUALITY

Table 6 presents qualitative comparisons of responses from the *Original*, *PrefixCache*, and *DualCache* methods for both an arithmetic prompt and an idiom-completion prompt. For the arithmetic question, all three methods correctly compute **3 bolts** through similar step-by-step reasoning, differing only in minor wording. For the idiom-completion task, all methods produce the standard completion and give accurate explanations of its meaning. These results indicate that different caching strategies preserve semantic fidelity and reasoning quality across both factual and conceptual queries, while primarily serving to improve efficiency without degrading interpretability or correctness.

Table 6: Qualitative comparison of responses across methods.

**Prompt:** A robe takes 2 bolts of blue fiber and half that much white fiber. How many bolts in total does it take?

| Original | PrefixCache | DualCache |
|---|---|---|
| The robe takes 2 bolts of blue fiber. It also takes half that much white fiber, so it takes 2/2 = 1 bolt of white fiber. In total, the robe takes 2 + 1 = 3 bolts of fiber. So, the value is 3 | The robe takes 2 bolts of blue fiber. It also takes half that much white fiber, so it takes 2/2 = 1 bolt of white fiber. In total, the robe takes 2 + 1 = 3 bolts of fiber. So, the value is 3 | The robe takes 2 bolts of blue fiber. It also takes half that much white fiber, so it takes 2/2 = 1 bolt of white fiber. In total, it takes 2 bolts + 1 bolt = 3 bolts of fiber. The final result is 3 |

**Prompt:** Finish the idiom and explain its meaning: A bird in the hand is worth...

| Original | PrefixCache | DualCache |
|---|---|---|
| A bird in the hand is worth two in the bush. This idiom means that having something certain and guaranteed is more valuable than the possibility of something that is uncertain or risky. It suggests that it's better to have a sure thing than to risk losing something valuable. The phrase comes from the idea that having a bird in one's hand is more secure than having two birds in the bush, which could be lost or harmed. | A bird in the hand is worth two in the bush. This idiom means that having something certain and guaranteed is more valuable than the possibility of gaining something more but is uncertain. It suggests that it's better to have a smaller amount of something that you know you have, rather than risking a larger amount of something that you don't have. The phrase is often used to encourage people to focus on what they already have and to avoid chasing after something that may not materialize. | A bird in the hand is worth two in the bush. This idiom means that having something certain and guaranteed is more valuable than the possibility of something greater but uncertain. It suggests that it's better to have a small, certain gain or reward rather than to risk a larger, potentially uncertain reward. In other words, it's better to be safe than sorry. |

Table 7: Qualitative comparison of responses with varying block size for DualCache.

**Prompt:** A robe takes 2 bolts of blue fiber and half that much white fiber. How many bolts in total does it take?

| Block Size 8 | Block Size 16 | Block Size 32 |
|---|---|---|
| The robe takes 2 bolts of blue fiber. It also takes half that much white fiber, so it takes 2/2 = 1 bolt of white fiber. In total, the robe takes 2 + 1 = 3 bolts of fiber. So, the value is 3 | The robe takes 2 bolts of blue fiber. It also takes half that much white fiber, so it takes 2/2 = 1 bolt of white fiber. In total, the robe takes 2 + 1 = 3 bolts of fiber. So, the value is 3 | The robe takes 2 bolts of blue fiber. It also takes half that much white fiber, so it takes 2/2 = 1 bolt of white fiber. In total, the robe takes 2 + 1 = 3 bolts of fiber. So, the value is 3 |

**Prompt:** Finish the idiom and explain its meaning: A bird in the hand is worth...

| Block Size 8 | Block Size 16 | Block Size 32 |
|---|---|---|
| A bird in the hand is worth two in the bush. This idiom means that having something certain and secure is more valuable than the possibility of gaining something more but is uncertain or risky. It emphasizes the importance of taking precautions and holding onto what you already have, rather than risking everything for a potential gain. | A bird in the hand is worth two in the bush. This idiom means that having something certain and secure is more valuable than the possibility of something that is uncertain or risky. It suggests that it's better to have a sure thing in your possession rather than taking a chance on something that may not work out. The phrase is often used to advise against taking unnecessary risks and to prioritize safety and security over potential gains. | A bird in the hand is worth two in the bush. This idiom means that having something certain and guaranteed is more valuable than the possibility of something greater but uncertain. It suggests that it's better to have a small, certain gain or reward rather than to risk a larger, potentially uncertain reward. In other words, it's better to be safe than sorry. |

Table 8: Qualitative comparison of responses under different threshold settings.

**Prompt:** A robe takes 2 bolts of blue fiber and half that much white fiber. How many bolts in total does it take?

| Threshold 0.7 | Threshold 0.8 | Threshold 0.9 |
|---|---|---|
| The robe takes 2 bolts of blue fiber. It also takes half that much white fiber, so it takes 2/2 = 1 bolt of white fiber. In total, it takes takes 2 + 1 = 3 bolts of fiber. So, the value is 3 **(NFE: 9)** | The robe takes 2 bolts of blue fiber. It also takes half that much white fiber, so it takes 2/2 = 1 bolt of white fiber. In total, the robe takes 2 + 1 = 3 bolts of fiber. So, the value is 3 **(NFE: 12)** | The robe takes 2 bolts of blue fiber. It also takes half that much white fiber, so it takes 2/2 = 1 bolt of white fiber. In total, the robe takes 2 + 1 = 3 bolts of fiber. So, the value is 3 **(NFE: 20)** |

**Prompt:** Finish the idiom and explain its meaning: A bird in the hand is worth...

| Threshold 0.7 | Threshold 0.8 | Threshold 0.9 |
|---|---|---|
| A bird in the hand is worth two in the bush. This idiom means that having something certain and guaranteed is more valuable than the possibility of something that is uncertain or risky. It suggests that it's better to have a sure thing than to risk losing something valuable. The phrase comes from the idea that having a bird in one's hand is more secure than having two birds in the bush, which could be lost or harmed. **(NFE: 74)** | A bird in the hand is worth two in the bush. This idiom means that having something certain and guaranteed is more valuable than the possibility of something that is uncertain or risky. It suggests that it's better to have a sure thing than to risk losing something valuable. The phrase comes from the idea that having a bird in one's hand is more secure than having two birds in the bush, which could be lost or harmed. **(NFE: 79)** | A bird in the hand is worth two in the bush. This idiom means that having something certain and guaranteed is more valuable than the possibility of something that is uncertain or risky. It suggests that it's better to have a sure thing than to risk losing something valuable. The phrase comes from the idea that having a bird in one's hand is more secure than having two birds in the bush, which could be lost or harmed. **(NFE: 86)** |

## B.2 EFFECT OF BLOCK SIZE IN DUALCACHE

Table 7 compares outputs from DualCache with varying block sizes (**8**, **16**, and **32**). Across both the arithmetic and idiom prompts, all configurations return identical answers, with consistent logical steps and explanatory clarity. No qualitative differences are observed in output content, suggesting that DualCache is robust to block size settings for these types of queries. This allows practitioners to adjust block size for computational efficiency without worrying about a loss of output quality.

## B.3 IMPACT OF DYNAMIC THRESHOLD SETTINGS

Table 8 explores the effect of varying dynamic threshold values (**0.7**, **0.8**, **0.9**). For the arithmetic and idiom tasks, all thresholds yield correct answers with coherent explanations, confirming that threshold tuning does not affect accuracy for straightforward prompts. However, higher thresholds noticeably increase the number of function evaluations (NFE), e.g., from **9** to **20** for the arithmetic task and from **74** to **86** for the idiom task. This suggests that threshold choice primarily influences computational cost, with quality remaining stable—enabling trade-offs between speed and resource usage for different application scenarios.

## B.4 MULTIMODAL GENERATION WITH `LLaDA-V`

To qualitatively analyze the effectiveness of our Fast-dLLM framework in multimodal scenarios, we conduct a visual case study where the model is tasked with generating a highly detailed image description. As illustrated in Figure 10, both the baseline model and our Fast-dLLM are given the same visual input and user prompt: "*Please describe the image in detail*."

The baseline model requires 63.0 seconds to complete the generation, producing a detailed and poetic description of the rural landscape. It highlights elements such as the weathered wooden barn, the soft pink sky, and the tranquil atmosphere.

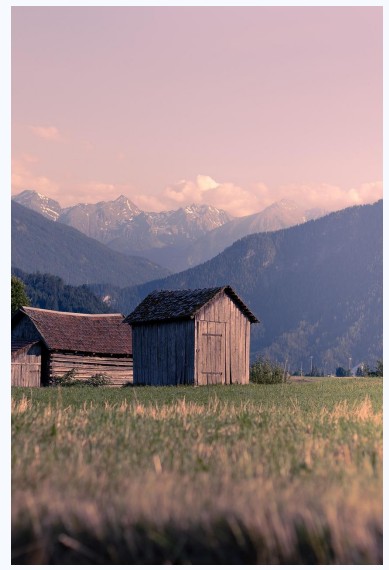

**User:** Please describe the image in detail.

**Baseline (63.0 secs):** The image captures a serene rural landscape. Dominating the scene is a wooden barn, its weathered exterior a testament to the passage of time. The barn is nestled in a field of green grass, which stretches out to meet the horizon. In the distance, a range of mountains rises, their peaks shrouded in a blanket of clouds. The sky above is a soft pink, suggesting either dawn or dusk. The perspective of the image is from a low angle, making the barn appear grand and imposing. The overall composition of the image creates a sense of depth and tranquility, inviting the viewer to imagine themselves in this peaceful rural setting.

**Fast-dLLM (6.8 secs):** The image captures a serene rural landscape. Dominating the scene is a wooden barn, its rustic charm enhanced by the gray shingles on its roof. The barn is nestled in a field of tall grass, which stretches out towards the horizon. The field is framed by a range of mountains, their peaks dusted with snow, adding a touch of tranquility to the scene. The sky above is a soft pink, suggesting either early morning or late afternoon. The perspective of the image is from a low angle, making the barn appear grand and imposing. The overall composition of the image creates a harmonious blend of man-made structures and the natural world.

Figure 6: Comparison between the baseline and Fast-dLLM on a visual description task. Fast-dLLM produces a comparable and faithful image caption in a fraction of the decoding time.

In contrast, our Fast-dLLM completes the task in just 6.8 seconds—a nearly $10\times$ speedup—while maintaining rich visual detail. It further enhances the description with additional grounding (e.g., "gray shingles on its roof", "touch of tranquility"), reflecting a strong alignment with both appearance and mood cues from the image. Notably, the generated caption retains compositional depth and stylistic fluency, illustrating the model's ability to balance fluency and factuality even under diffusion-based parallel decoding.

This case highlights how LLAda-V with Fast-dLLM decoding enables high-quality vision-language generation at significantly improved efficiency, paving the way for faster and more interactive multimodal applications.

## C  EXPERIMENT DETAILS

### C.1  FURTHER EXPERIMENTS WITH LLaDA-V

Table 9: Effect of block length on performance (MathVista, 48 Steps)

| Block Length | 4 | 8 | 16 | 32 | 96 |
|---|---|---|---|---|---|
| Accuracy (%) | 51.2 | 50.7 | 51.8 | 52.3 | 59.7 |
| Throughput (tok./s) | 6.1 | 6.2 | 5.5 | 5.5 | 5.6 |

Table 10: MathVista Performance with Fast-dLLM at different refresh intervals (block length = 96)

| Refresh Interval | 2 | 4 | 8 | 16 | 32 |
|---|---|---|---|---|---|
| Accuracy (%) | 59.2 | 59.2 | 58.2 | 57.1 | 56.6 |
| Throughput (tok./s) | 15.9 | 19.5 | 21.1 | 25.2 | 28.2 |

In Table 9, we investigate how the choice of block length affects the performance of LLaDA-V on MathVista under a fixed decoding length of 48 steps. The results show that the model achieves the highest accuracy with a block length of 96. However, when reducing the block size to 8 or 4, the accuracy drops significantly by over 8%.

Given this sensitivity to block length, we choose not to break the output into small blocks for updating caches individually. Instead, we keep the block length fixed at 96 and adopt a refresh-based strategy: the cache is updated only every $r$ decoding steps using the most recent full block. As shown in Table 10, increasing the refresh interval leads to consistent gains in throughput—from 15.9 tokens/s at interval 2 to 28.2 tokens/s at interval 32. While accuracy drops slightly with larger intervals, it remains above 56.6%, suggesting that aggressive refresh scheduling can yield substantial speedups with only minor performance degradation.

## C.2 PERFORMANCE COMPARISON BETWEEN THRESHOLD AND FACTOR STRATEGY

Table 11: Performance comparison between **Threshold** and **Factor** confidence-aware decoding on GSM8K and MATH benchmarks with generation lengths of 256 and 512. Each block shows accuracy (top row) and throughput with speedup (bottom row). Factor decoding provides favorable trade-offs in most settings.

| Benchmark | Gen. Len | Threshold | Factor |
|---|---|---|---|
| GSM8K (5-shot) | 256 | 78.5 | 77.5 |
| | | 54.4 (8.1×) | 78.5 (11.7x) |
| | 512 | 77.2 | 74.8 |
| | | 35.3 (11.0×) | 47.1 (14.7x) |
| MATH (4-shot) | 256 | 33.2 | 32.0 |
| | | 51.7 (5.7×) | 78.3 (8.6x) |
| | 512 | 36.0 | 35.2 |
| | | 47.1 (5.9×) | 64.6 (8.1x) |

We compare the performance of our threshold-based and factor-based confidence-aware parallel decoding strategies on GSM8K and MATH benchmarks (Table 11). While the threshold strategy achieves marginally better accuracy in most settings (e.g., 78.5% vs. 77.5% on GSM8K with 256 tokens), the factor strategy demonstrates substantially superior throughput performance.

Specifically, factor decoding achieves 1.4-1.5× higher throughput than threshold decoding across all settings. On GSM8K with 256 tokens, factor decoding reaches 78.5 tokens/sec (11.7× speedup) compared to 54.4 tokens/sec (8.1× speedup) for threshold decoding. This throughput advantage becomes even more pronounced on longer generation tasks—for GSM8K with 512 tokens, factor decoding attains 47.1 tokens/sec while threshold only achieves 35.3 tokens/sec.

The results demonstrate that factor decoding offers a compelling trade-off: it sacrifices minimal accuracy (typically 1-3%) in exchange for significant throughput improvements (40-50% higher). This makes factor decoding particularly attractive for latency-sensitive applications where the slight accuracy reduction is acceptable. The consistent pattern across both benchmarks and generation lengths validates the robustness of the factor strategy's theoretical foundation, which adaptively controls parallelism based on the confidence bound $(n + 1)\epsilon < f$.

## C.3 COMPARISON BETWEEN LLADA AND LLADA-1.5

We compare the performance of LLaDA and its enhanced version LLaDA-1.5 across both GSM8K (5-shot) and MATH (4-shot) benchmarks under two generation length settings (256 and 512 tokens), as shown in Table 12. Each cell reports accuracy and decoding throughput (in tokens per second), along with the relative speedup over the greedy baseline.

Across GSM8K settings, LLaDA-1.5 consistently improves accuracy over the original LLaDA, achieving a notable +2.2% absolute gain at 256-token generation and +3.2% at 512-token generation. Furthermore, it maintains strong decoding efficiency, with throughput reaching 59.4 tokens/sec at 256 tokens, improving upon LLaDA's 54.1 tokens/sec under the same setting.

On the MATH benchmark, accuracy between the two versions remains comparable. However, LLaDA-1.5 slightly improves throughput at 256 tokens (53.7 vs. 51.7) while incurring a mild

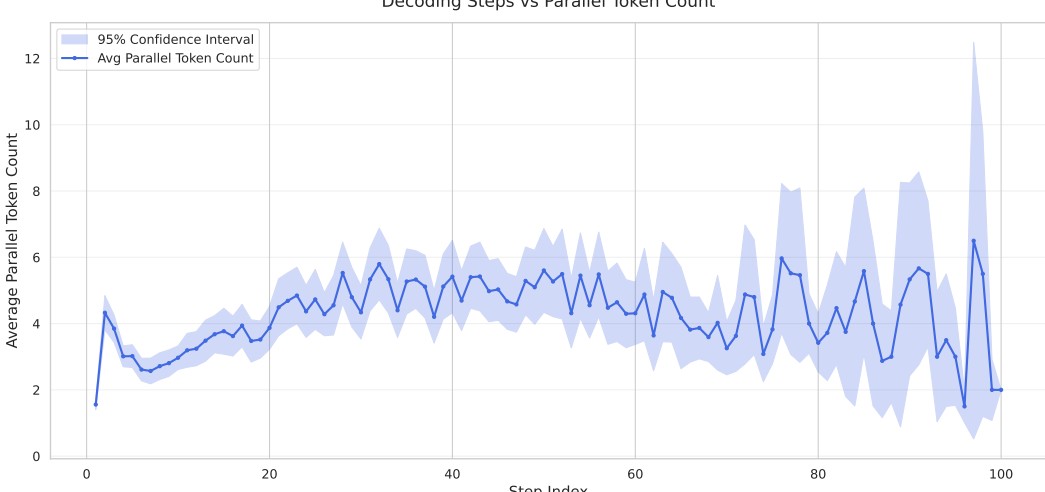

Figure 7: Average number of tokens generated at each decoding step. Blue line shows the mean token count, and the shaded area denotes the 95% confidence interval.

Table 12: Performance comparison between LLaDA and LLaDA-1.5. Each cell presents the accuracy and the decoding throughput in tokens per second with relative speedup to the LLaDA baseline (bottom row, blue: tokens per second/orange: relative speedup).

| Benchmark | Gen Length | LLaDA (Fast-dLLM) | LLaDA 1.5 (Fast-dLLM) |
|---|---|---|---|
| GSM8K (5-shot) | 256 | 78.5 | 80.7 |
| | | 54.1 (8.1×) | 59.4 (8.9×) |
| | 512 | 77.2 | 80.4 |
| | | 35.3 (11.0×) | 33.0 (10.3×) |
| MATH (4-shot) | 256 | 33.2 | 32.6 |
| | | 51.7 (5.7×) | 53.7 (5.9×) |
| | 512 | 36.0 | 35.1 |
| | | 47.1 (5.9×) | 41.1 (5.1×) |

efficiency regression at the 512-token setting (41.1 vs. 47.1). This suggests that while LLaDA-1.5 introduces enhancements beneficial for shorter or moderate decoding contexts, longer sequences may require further optimization.

Overall, LLaDA-1.5 consistently provides either superior accuracy or better decoding speed across settings, demonstrating better performance-efficiency trade-offs and highlighting the benefit of incorporating adaptive improvements on top of the base LLaDA architecture.

## C.4    ANALYSIS OF PARALLEL TOKEN COUNTS ACROSS DECODING STEPS

To better understand the behavior of factor-based parallel generation, we analyze the average number of tokens generated at each decoding step. Specifically, we collect statistics from all intermediate steps of the sampling process and compute the average number of tokens generated in parallel per step. The results are visualized in Figure 7, along with a 95% confidence interval indicating cross-sample variability.

As shown in Figure 7, the average number of tokens generated in parallel gradually increases during the early to middle stages of decoding, peaking roughly between step 30 to step 60. After this peak, the parallelism tends to slightly decline toward the end of generation. This suggests that the model becomes more confident in generating outputs during the mid-decoding phase, allowing it to produce more tokens simultaneously. Toward the final steps, the decoding process tends to become more conservative, reducing the number of tokens produced at each step.

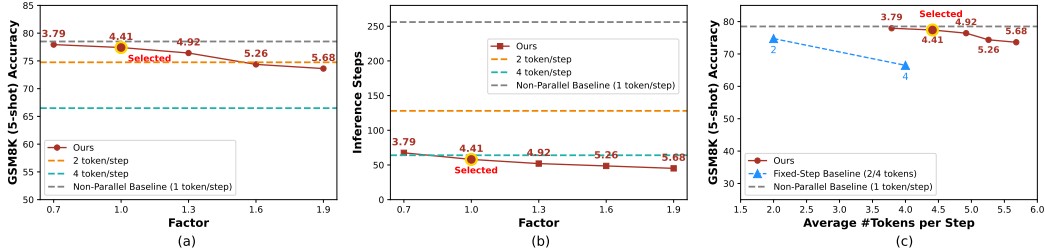

Figure 8: (a) GSM8K (5-shot) accuracy across different factor values using our factor-based decoding strategy. Numbers above each point indicate the average number of tokens decoded per step. The dashed lines show the accuracy of the baseline method with 2 or 4 tokens per step, and the non-parallel (1 token/step) baseline. (b) The corresponding number of inference steps needed under each factor setting. Our method generally requires significantly fewer steps than fixed-step baselines. (c) Accuracy versus average number of tokens decoded per step on GSM8K (5-shot). Our factor-based decoding achieves better accuracy-efficiency trade-offs compared to baselines. The red "Selected" point represents the setting chosen in our main results.

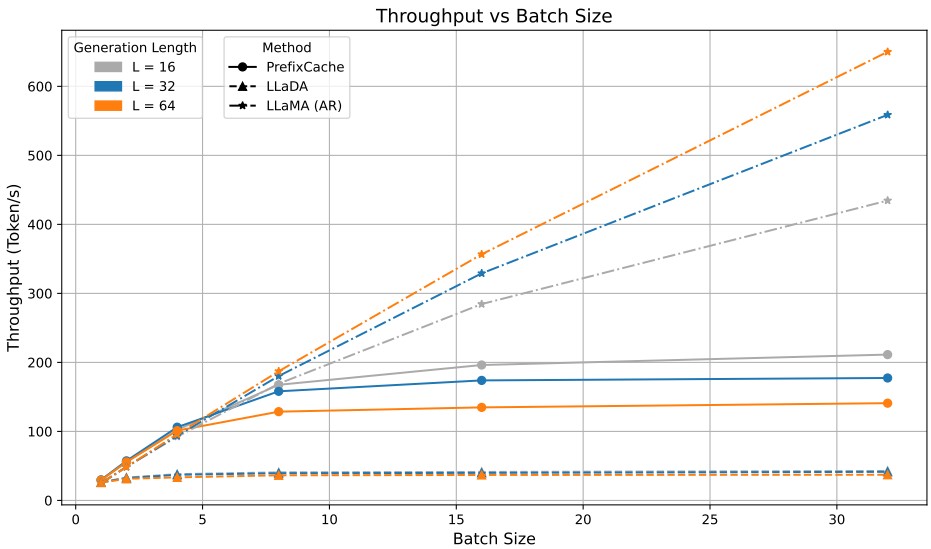

Figure 9: Throughput comparison between **PrefixCache**, **LLaDA**, and **LLaMA** under different generation lengths and batch sizes. All models are evaluated on an **NVIDIA A100** GPU with the prefill length fixed at 256.

The shaded confidence interval reveals greater variance in later decoding steps, indicating instability and inconsistent generation behavior across samples. This is expected since tail-end decoding steps tend to handle only a few remaining tokens required to complete the output, and the number of remaining tokens could differ widely among different samples (e.g., due to early completion or padding).

These observations are important for understanding how decoding efficiency can be optimized: increasing parallelism during high-confidence phases (middle steps) offers computational savings, while conservative behavior near boundaries maintains quality.

## C.5   THROUGHPUT COMPARISON UNDER VARYING BATCH SIZES

All experiments are conducted on an NVIDIA A100 GPU, with the prefill length fixed to 256 tokens. The generation length is varied among 16, 32, and 64 tokens, and batch sizes range from 1 to 32. This setup reflects realistic deployment scenarios, allowing the evaluation of decoding efficiency under diverse conditions.

It should be noted that parallel decoding allows multiple tokens to be generated simultaneously affected by dummy input tokens. To ensure fairness, we focus solely on the acceleration provided by caching techniques.

PrefixCache is designed as an acceleration mechanism for LLaDA, a diffusion-based LLM, and successfully boosts the throughput significantly. Figure 9 shows that **PrefixCache achieves consistent improvements across all batch sizes and generation lengths**, making it particularly suited for scenarios with smaller generation lengths and larger batch sizes. For instance, with a generation length of 16 and batch size of 32, PrefixCache achieves a throughput of over 211 tokens/s, significantly outperforming the native LLaDA which reaches only 43 tokens/s, demonstrating nearly $5\times$ improvement.

While LLaDA exhibits limited scalability with increasing batch sizes—its throughput plateaus after batch size 8—this limitation is inherent to diffusion-based LLMs, which are compute-bound by nature. In contrast, LLaMA, an autoregressive (AR) model, benefits greatly from large batch sizes. As the batch size increases, LLaMA shifts from being memory-bound to compute-bound, allowing it to achieve high absolute throughput at larger batch settings.

These results highlight the practical advantages of PrefixCache in accelerating compute-bound diffusion models like LLaDA, especially for latency-critical and high-throughput applications. Furthermore, the scalability and efficiency provided by PrefixCache bridge the gap between diffusion-based LLMs and AR models like LLaMA, showcasing its importance for large-scale deployment settings.

## D RELATED WORK

### D.1 DIFFUSION LLM

Diffusion models have emerged as a transformative paradigm in generative modeling, initially achieving remarkable success in continuous domains such as image (Rombach et al., 2022; Nichol et al., 2022; Ramesh et al., 2021; Saharia et al., 2022) and audio synthesis (Yang et al., 2023; Huang et al., 2023) before expanding into natural language processing. Recent advancements in discrete diffusion models (Austin et al., 2021; Nie et al., 2025a;b; Hoogeboom et al., 2021; Campbell et al., 2022; He et al., 2022; Meng et al., 2022; Reid et al., 2022; Sun et al., 2022; Kitouni et al., 2023; Zheng et al., 2023; Chen et al., 2023; Ye et al., 2023; Sahoo et al., 2024; Shi et al., 2024; Zheng et al., 2024; Gat et al., 2024; Yu et al., 2025b;a) have reshaped the landscape of text generation, offering a viable alternative to autoregressive (AR) paradigms in large language models (LLMs). These models address the inherent challenges of discrete data by redefining noise injection and denoising processes through innovative mathematical formulations.

**Theoretical Foundations of Discrete Diffusion** Diffusion models for discrete data were first explored in (Sohl-Dickstein et al., 2015; Hoogeboom et al., 2021). Subsequently, D3PM (Austin et al., 2021) provided a more general framework. This framework models the forward noising process as a discrete state Markov chain using specific transition matrices. For the reverse process, D3PM learns a parameterized model of the conditional probability of the original data given a noised version by maximizing the Evidence Lower Bound (ELBO). CTMC (Campbell et al., 2022) further extended D3PM to a continuous-time setting, formalizing it as a continuous-time Markov Chain (CTMC). In a distinct approach, SEDD (Lou et al., 2023) learns the reverse process by parameterizing the ratio of marginal likelihoods for different data instances at a given noising timestep. This ratio model is then trained using a Denoising Score Entropy objective. More recently, research on Masked Diffusion Models (MDMs) by MDLM (Shi et al., 2024; Sahoo et al., 2024; Zheng et al., 2024) and RADD (Ou et al., 2024) has introduced significant clarifications. These studies have demonstrated that different parameterizations of MDMs can be equivalent.

**Integration with Pre-trained Language Models** A critical breakthrough involves combining discrete diffusion with existing LLM architectures. Diffusion-NAT (Zhou et al., 2023) unifies the denoising process of discrete diffusion with BART's (Lewis et al., 2019) non-autoregressive decoding, enabling iterative refinement of masked tokens. By aligning BART's inference with diffusion steps, this approach leverages pre-trained knowledge while maintaining generation speed 20× faster than comparable AR transformers. Similarly, the LLaDA (Nie et al., 2025b) and DiffuLLaMA (Gong et al., 2024) framework scales diffusion to 7B parameters using masked denoising, while LLaDA

and Dream (Ye et al., 2025) demonstrating competitive performance with autoregressive baselines like LLaMA3 (Grattafiori et al., 2024) through recursive token prediction across diffusion timesteps. Beyond text-only generation, Lumina-DiMOO (Xin et al., 2025) extends discrete diffusion modeling to multimodal tasks, achieving state-of-the-art performance on text-to-image generation and image understanding by utilizing fully discrete diffusion across modalities with higher sampling efficiency than hybrid AR-Diffusion paradigms.

**One-Step and Few-Step Diffusion Models** Recent work has focused on reducing the number of denoising steps required for high-quality generation. OneFlowSeq (Anonymous, 2025) distills multi-step diffusion teachers into one-step generators through MeanFlow-based supervision and Jacobian-vector product signals, achieving state-of-the-art Seq2Seq performance with significantly reduced inference time while requiring fewer trainable parameters. dParallel (Chen et al., 2025b) introduces certainty-forcing distillation that trains diffusion models to achieve high certainty on masked tokens more rapidly and in parallel, reducing decoding steps from 256 to 30 on GSM8K with 8.5× speedup. Learning to Parallel (Learn2PD) (Bao et al., 2025) proposes a lightweight adaptive filter model that predicts whether each token position matches the final output, achieving up to 22.58× speedup on LLaDA without performance degradation, and up to 57.51× when combined with KV-Cache.

## D.2 LLM ACCELERATION

**Key-Value Cache.** Key-Value (KV) Cache is a fundamental optimization technique in modern large language model (LLM) inference with Transformer architecture (Vaswani, 2017). It enables efficient autoregressive text generation by storing and reusing previously computed attention states. However, it is non-trivial to apply KV Cache in diffusion language models such as LLaDA due to full attention. Block diffusion (Arriola et al., 2025) overcomes key limitation of previous diffusion language models by generating block-by-block so that key and values of previously decoded blocks can be stored and reused. Several recent works have proposed specialized caching mechanisms for diffusion LLMs. dLLM-Cache (Liu et al., 2025) introduces a training-free adaptive caching framework that combines long-interval prompt caching with partial response updates guided by feature similarity, achieving up to 9.1× speedup. Sparse-dLLM (Song et al., 2025) proposes the first training-free dynamic cache eviction method featuring delayed bidirectional sparse caching, leveraging the observation that pivotal tokens remain salient across decoding steps while low-relevance tokens stay unimportant, achieving up to 10× higher throughput. DPad (Chen et al., 2025a) restricts attention to a structured subset of suffix tokens through a sliding window and distance-decay dropout strategy, delivering up to 61.4× speedup while maintaining comparable accuracy.

**Non-Autoregressive Generation** Non-autoregressive (NAR) generation marks a fundamental shift from sequential token generation by enabling the simultaneous generation of multiple tokens, significantly accelerating inference (Xiao et al., 2023). Initially introduced for neural machine translation, NAR methods have since been extended to a variety of tasks, including grammatical error correction, text summarization, dialogue systems, and automatic speech recognition. Although NAR generation offers substantial speed advantages over autoregressive approaches, it often sacrifices generation quality. Diffusion LLMs represent a recent paradigm for non-autoregressive text generation; however, prior work (Nie et al., 2025b) has struggled to realize the expected acceleration due to a notable drop in output quality.

## E LLM USAGE

During manuscript preparation, we used large language models —strictly for language polishing of paragraphs and sentences (grammar, flow, and tone). These tools were not used to generate ideas, design experiments, or determine conclusions. All technical content, methodology, and interpretations were written, verified, and approved by the authors. To reduce risks of factual drift or citation errors, we required human review of every model-edited sentence and cross-checked all references against primary sources. The authors take full responsibility for the accuracy and integrity of the manuscript.

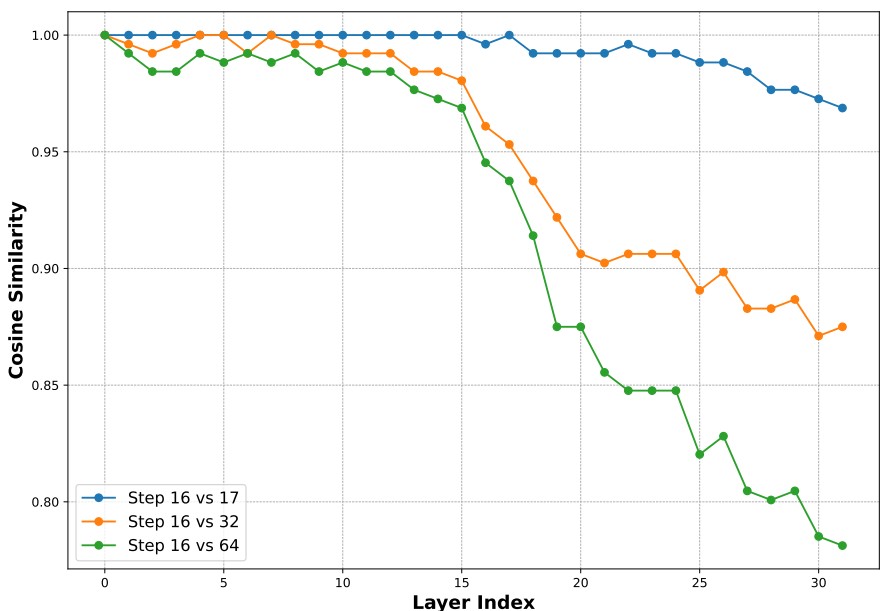

Figure 10: KV cache similarity across all layers between adjacent diffusion steps.

## F  KV ACTIVATION SIMILARITY

As shown in the figure, adjacent diffusion steps (e.g., step 16 vs. 17) consistently exhibit high cosine similarity across all layers, indicating that our caching mechanism is well-justified for short-term reuse. However, the similarity decreases in deeper layers as step intervals widen (e.g., step 16 vs. 64), suggesting that KV activations evolve more rapidly toward the top of the network. This non-uniformity across layers and steps points to opportunities for improvement. In future work, we plan to explore layer-aware or adaptive caching strategies that more precisely account for per-layer similarity patterns.

## G  DPARALLEL COMBINED WITH FAST-DLLM

Our confidence-aware decoding strategy remains effective when applied to distilled few-step diffusion models like dParallel, which naturally exhibit higher confidence due to distillation. Using a threshold of 0.5, dParallel combined with our decoding method already achieves large speedups. Moreover, by further integrating our KV cache, we observed an additional 1.84× throughput gain (from 49.9 to 92.0 tokens/s) with a slight accuracy improvement (from 76.5% to 77.1%) on GSM8K (see table above). The reduced number of decoding steps per block in dParallel improves cache reuse fidelity, making our caching design even more precise. As for One-Step Diffusion LLM, while it explores aggressive timestep reduction, its evaluation focuses primarily on perplexity rather than downstream tasks, so we could not include it in our setting. Exploring how our techniques generalize to extreme distillation regimes remains a valuable direction for future work.

Table 13: Performance comparison of LLaDA baseline, dParallel (threshold 0.5), and dParallel with prefix caching (threshold 0.5) on GSM8K in terms of throughput (tokens/s) and accuracy (%). The dParallel model benefits from distillation-induced higher confidence, enabling effective application of confidence-aware decoding. Integrating KV cache further boosts throughput by 1.84× with a slight accuracy gain.

| Model | Throughput (tokens/s) | Accuracy (%) |
|---|---|---|
| Baseline (LLaDA) | 6.7 | 79.3 |
| dParallel (Threshold 0.5) | 49.9 | 76.5 |
| dParallel + Prefix Cache (Threshold 0.5) | 92.0 | 77.1 |

## H  MEMORY USAGE

While our block-wise KV caching mechanism introduces changes in memory access patterns, it does not significantly increase memory usage. As shown in the table below, both PrefixCache and DualCache reuse KV entries from a single full-sequence cache, split across blocks rather than duplicated. For instance, in the 256+256 case, DualCache reduces computation by over 16× (7.68T → 0.48T FLOPs), while memory only increases slightly (15.07G → 15.36G). Even at 1024+1024, VRAM remains well within 16.6GB, showing practicality for large-scale input. PrefixCache reuses only the prefix, while DualCache retains all but the current decoding block, providing more effective reuse at minimal extra memory cost. Overall, our method achieves substantial speedup with marginal VRAM overhead, and scales well to long sequences and large models.

Table 14: FLOP count (T) and peak VRAM usage (G) under different input and generation sequence lengths (256+256, 512+512, 1024+1024) for LLaDA baseline, PrefixCache, and DualCache. DualCache achieves over 16× reduction in computation (7.68T → 0.48T FLOPs) with only marginal memory increase (15.07G → 15.36G) in the 256+256 case, demonstrating high computational efficiency and memory scalability.

| Method | 256+256 | | 512+512 | | 1024+1024 | |
|---|---|---|---|---|---|---|
| | FLOPs (T) | VRAM (G) | FLOPs (T) | VRAM (G) | FLOPs (T) | VRAM (G) |
| LLaDA (Baseline) | 7.68 | 15.07 | 15.36 | 15.20 | 30.71 | 15.44 |
| PrefixCache | 3.84 | 15.63 | 7.68 | 16.31 | 15.36 | 17.69 |
| DualCache | 0.48 | 15.36 | 0.48 | 15.76 | 0.48 | 16.57 |

## I  COMPARE WITH RELATED WORKS

While dLLM-Cache focuses on content-based adaptive KV caching using value similarity, our block-wise KV cache relies on temporal similarity across adjacent steps, offering a complementary approach. DPad targets a different axis by pruning redundant suffix tokens to reduce attention cost, whereas our method optimizes the decoding process itself via caching and confidence-aware parallel decoding. Notably, both methods are compatible with our approach, and can be combined for multiplicative speedups, as acknowledged in the DPad paper. We see these works as complementary, addressing different bottlenecks in dLLM inference, and appreciate the opportunity to position Fast-dLLM within this evolving landscape.

Table 15: Accuracy (%) and throughput (tokens/s) on GSM8K 5-shot for 256- and 512-token sequences. Fast-dLLM denotes our full system (confidence-aware parallel decoding + block-wise KV cache). dLLM-Cache adopts value-similarity-based adaptive caching, whereas DPad prunes redundant suffix tokens. Our approach is orthogonal and can be combined with others for multiplicative gains (e.g., dPad+Parallel+PrefixCache reaches 54.0 tokens/s at 256 length with 77.6% accuracy).

| Method | 256 tokens | | 512 tokens | |
|---|---|---|---|---|
| | Acc (%) | Thru (tok/s) | Acc (%) | Thru (tok/s) |
| dLLM-Cache | 79.2 | 18.0 | 78.0 | 9.2 |
| DPad | 79.1 | 7.3 | 80.1 | 7.2 |
| + Parallel | 80.5 | 20.8 | 81.0 | 21.2 |
| + Parallel + PrefixCache | 77.6 | 54.0 | 79.2 | 53.8 |
| Fast-dLLM (ours) | 78.5 | 54.4 | 77.3 | 35.3 |
| PrefixCache | 79.5 | 21.2 | 77.0 | 10.4 |
| Parallel | 79.2 | 16.5 | 77.6 | 18.6 |

