# OpenReview forum: "Fast-dLLM: Training-free Acceleration of Diffusion LLM by Enabling KV Cache and Parallel Decoding"
_ICLR.cc/2026/Conference — ICLR 2026 Poster_

### Official Review · Reviewer_MBqz · 2025-10-30

**Soundness:** 3
**Presentation:** 3
**Contribution:** 3
**Rating:** 6
**Confidence:** 4

**Summary:**

This paper investigates the slow inference speed of diffusion-based large language models, stemming from the lack of Key-Value (KV) cache support and quality degradation in parallel decoding. The proposed Fast-dLLM method introduces a block-wise approximate KV cache tailored for bidirectional attention and a confidence-aware parallel decoding strategy. This strategy dynamically selects tokens above a confidence threshold, enabling efficient cache reuse and safe, training-free parallel token generation.

**Strengths:**

1. The method is training-free, making it an easily applicable plug-in for compatible diffusion language models.
2. It includes a comprehensive set of ablation studies analyzing the impact of key components and hyperparameters, such as cache block size, confidence threshold, cache variants, and generation/prefill length.
3. The empirical results are significant, showing marked improvements in throughput across multiple benchmarks with minimal accuracy loss.

**Weaknesses:**

1. The mechanism of 'DualCache' is not clearly explained; it is unclear how caching suffix tokens saves computation and accelerates inference. Additionally, more details on the extra memory overhead introduced by the KV cache need to be disclosed.  In addition, the paper claims that the cache update introduces “negligible overhead” in Figure 2, but it provides no concrete explanation or timing comparison.

2. The novelty of the block-wise KV cache contribution is unclear. The manuscript fails to sufficiently differentiate its caching method from existing work like Block-diffusion, and the experimental section lacks relevant diffusion model baselines.

3. The reported speedup ratios raise concerns about potential metric inflation. Furthermore, the datasets used are primarily focused on math and code problems, lacking experimental data on benchmarks specifically designed for evaluating inference acceleration.

4. The conceptual relationship and practical distinction between threshold- and factor-based strategies are insufficiently clarified. It is unclear whether the factor-based rule is a relaxed, adaptive, or independent variant, or under which conditions one should be preferred. A clearer rationale linking these strategies would improve comprehension and applicability.

5. While the paper qualitatively illustrates KV similarity, it does not provide quantitative measures showing how approximation errors change with block size, decoding length, or modality, making it difficult to assess cache mismatch risks. It would be helpful to report KV similarity decay curves across decoding steps for different block sizes and tasks, correlate these similarities with downstream task metrics (e.g., accuracy, EM, BLEU) to establish a “similarity threshold → refresh” rule.

6. The assumption of a “well-defined joint PMF with self-consistent marginals” may not hold for real diffusion LLMs trained approximately. The implications of this idealization are not discussed in detail, limiting interpretability of theoretical guarantees.

7. The paper does not include a comparison with autoregressive (AR) models. As a result, it remains unclear whether the diffusion + Fast-dLLM approach is competitive with state-of-the-art AR systems in realistic serving scenarios.

**Questions:**

1. The paper does not clarify how to automatically tune the commonly used 0.9 confidence threshold across different model scales, temperatures, or tasks, nor whether a simple calibration method (e.g., based on confidence–accuracy curves) can be applied. It also leaves open how to select the factor-based hyperparameter \(f\), and whether a universal default exists or task-specific tuning is required.

2. The paper does not quantify the computational cost (FLOPs or wall-clock time) of recomputing all blocks after completing one, nor does it explore whether incremental updates—such as refreshing only neighboring blocks or subsets of Keys/Values—could achieve comparable accuracy more efficiently. Profiling or throughput measurements would help clarify these trade-offs.

3. In the MathVista experiments, Fast-dLLM exhibited a noticeable performance degradation. Could you give a detailed analysis?

---

> ### Author Response · Authors · 2025-11-21
> **Response to Reviewer MBqz (1/3)**
>
> ## Q1:  The mechanism of 'DualCache' and memory overhead
> ## A1:
> DualCache works by reusing KV activations for all but the current decoding block, allowing us to avoid recomputing attention over long suffixes at each step. While refreshing the cache requires feeding the full sequence into the model, this is consistent with the baseline behavior in masked diffusion LLMs, which also recompute over the full context each step. Thus, the claimed “negligible overhead” refers to no additional cost relative to the baseline—not absolute cost. As shown in the table, DualCache reduces FLOPs by over 16× (7.68T → 0.48T) with only ~0.3G VRAM increase compared to baseline. We will clarify this point in the main text and table caption to avoid confusion. The test is conducted in A100-80G GPU and 256 256 represents the prefix length and generation length is equal to 256. Here we evaluate this exact step's flops and memory cost.
>
> |                     | 256 256           | 512 512           | 1024 1024          |
> |---------------------|-------------------|-------------------|--------------------|
> | LLaDA (Baseline)    | 7.68T/15.07G      | 15.36T/15.20G     | 30.71T/15.44G      |
> | PrefixCache         | 3.84T/15.63G      | 7.68T/16.31G      | 15.36T/17.69G      |
> | DualCache           | 0.48T/15.36G      | 0.48T/15.76G      | 0.48T/16.57G       |
>
> ## Q2: Relation between Block Diffusion
> ## A2:
> While both our work and Block Diffusion employ "block-wise" concepts, they address fundamentally different problems. Block Diffusion proposes a new hybrid autoregressive-diffusion architecture that requires specialized training, whereas our method targets existing bidirectional masked diffusion LLMs and uses a training-free block-wise KV cache designed specifically for temporal redundancy across diffusion steps. Additionally, our work operates at a much larger scale (7–8B models) and focuses on challenging downstream tasks (e.g., GSM8K, HumanEval), unlike Block Diffusion which is evaluated on small-scale LM benchmarks. Thus, our cache mechanism is novel in its design goals, application setting, and evaluation scope, and complements rather than replicates prior work.
>
> ## Q3: More benchmark results
> ## A3:
> Thank you for raising this concern. The reported 27.6× speedup refers to a specific long-sequence setting (e.g., 8-shot for GSM8K, generation length = 1024) designed to stress-test performance under extreme latency conditions. To avoid misleading impressions, we also report more typical benchmark scenarios in Table 1 and Table 2, covering moderate-length math and code tasks.
>
> Beyond these, we include results on IFEval, a general-purpose instruction-following benchmark, demonstrating that Fast-dLLM achieves strong speedups (e.g., 3.7× at length 512) with minimal accuracy drop, indicating solid generalization beyond math/code.
>
> We note that the diffusion LLM community currently lacks standardized benchmarks for inference acceleration. As a result, we report speedups on widely used benchmarks, which we believe are representative of real-world usage.
>
> | Benchmark    | Metric | LLaDA Baseline | +Cache        | +Parallel     | +Cache+Parallel (Fast-dLLM) |
> |--------------|------------|----------------|---------------|---------------|-----------------------------|
> | IFEval 256      | Accuracy        | 57.1            | 53.6          | 57.1          | 54.0                        |
> |              | Throughput          | 15.7            | 19.7          | 30.6          | 36.0                        |
> | IFEval 512      | Accuracy        | 58.0            | 57.5          | 57.9          | 57.5                        |
> |              | Throughput          | 8.2             | 12.0          | 22.2          | 30.0                        |

---

> ### Author Response · Authors · 2025-11-21
> **Response to Reviewer MBqz (2/3)**
>
> ## Q4: Conceptual relationship and practical distinction between threshold- and factor-based strategies
> ## A4:
> Thank you for the insightful question. The threshold-based and factor-based strategies are conceptually related—they both aim to dynamically determine how many tokens to decode per step based on model confidence—but differ in how the confidence signal is interpreted and operationalized.
> - The threshold strategy sets a fixed confidence level (e.g., 0.9) and decodes all tokens whose confidence exceeds that value. It makes conservative, high-confidence updates.
> - The factor strategy, by contrast, adapts the decoding width based on a simple constraint: for each step, we decode the top-k tokens as long as their lowest confidence c satisfies the condition k × (1 − c) < f, where f is a tunable factor. This strategy enables adaptive parallelism—larger k when confidence is high—and tends to yield higher throughput at the cost of a small accuracy drop. It can be viewed as a relaxed and adaptive variant of the threshold method, focusing on bounding cumulative uncertainty rather than enforcing a hard minimum.
>
> Practically, the factor strategy acts as a more relaxed version of the threshold rule, allowing greater decoding parallelism at the cost of slightly lower accuracy (as shown in Appendix C.2). The threshold rule is preferred when accuracy preservation is paramount and a clear decision boundary is desired; the factor rule is suitable when throughput is prioritized and modest accuracy trade-offs are acceptable.
>
> ## Q5: Quantitative analysis of KV similarity
> ## A5:
> While we qualitatively illustrate KV similarity in the main text, its practical effects are reflected in Figure 4, where different block sizes (i.e., cache reuse durations) show how accuracy degrades as cache staleness increases—highlighting the risk of cache mismatch with longer reuse. For example, beyond block size 64, we observe a notable accuracy drop, indicating similarity decay over time.
>
> Related work such as dLLM-Cache [arXiv:2506.06295] has conducted similarity-based KV cache evaluation using value-wise cosine similarity to guide adaptive updates. Their methodology is complementary to ours and could be used to develop refresh policies based on real-time similarity signals (e.g., refresh when similarity < τ). We see this as a promising direction for future integration, and we also cite dLLM-Cache in the revised version.
>
> ## Q6: The assumption of a “well-defined joint PMF with self-consistent marginals”
> ## A6:
> In practice, while MDM may not strictly satisfy this property, its behavior typically offers a close approximation. As MDMs become larger and more powerful, their learned distributions will better approximate such consistency. One of our main contributions is demonstrating that, guided by this theory (namely, our Confidence-Aware Parallel Decoding strategy), we achieve significant success on realistic, imperfectly trained MDMs, such as LLaDA and Dream.
>
> ## Q7: Comparison with AR models
> ## A7:
> We do include a comparison with AR models in Figure 1(a), specifically evaluating LLaMA-3 8B and Qwen 2.5-7B, two competitive AR baselines. Under consistent settings (GSM8K 5-shot, batch size = 1), Fast-dLLM (e.g., LLaDA+KVCache+Parallel) achieves comparable or higher accuracy while offering significantly higher throughput—up to 8.1× faster than vanilla LLaDA and outperforming AR models in speed. This demonstrates that diffusion + Fast-dLLM is competitive with AR models in realistic serving scenarios.

---

> ### Author Response · Authors · 2025-11-21
> **Response to Reviewer MBqz (3/3)**
>
> ## Q8: Calibration of hyperparameters
> ## A8:
> Thank you for the thoughtful question. In our experiments, we used a fixed threshold of 0.9 and a factor value of 1.0 across all tasks and model scales, and found them to work robustly without task-specific tuning. This suggests that both strategies are relatively stable under default settings within the diffusion LLM context.
>
> That said, we agree that calibrating these hyperparameters—especially under varying generation temperatures, model sizes, or domains—is an important direction. For the threshold strategy, one possible approach is to fit confidence–accuracy curves on a small validation set to select an appropriate threshold that matches a desired precision level. For the factor-based method, the parameter f can similarly be tuned by grid search or optimized based on speed–accuracy trade-off curves.
>
> ## Q9: Quantitative analysis of the computational cost (FLOPs) and partial refresh
> ## A9:
> Thank you for the thoughtful question. As shown in the table, DualCache reduces FLOPs by over 16× compared to full-step decoding (e.g., 7.68T → 0.48T for 256+256), even though cache refresh requires feeding the full sequence once per block. Since baseline diffusion LLMs already recompute over the full sequence at each step, our block-wise refresh introduces no extra computational overhead relative to the default setting, and thus offers significant end-to-end savings.
>
> Regarding incremental update strategies, we agree that selectively refreshing parts of the cache could offer further gains. Recent work such as Sparse-dLLM [arXiv:2508.02558], which builds on Fast-dLLM, explores this direction by introducing dynamic cache eviction with sparse attention, retaining only salient tokens. This method achieves up to 10× throughput with comparable accuracy and lower memory usage. We view such fine-grained updates as a promising future direction and complementary to our current design. We also add this paper to the related work to provide a more complete view of the ongoing research landscape.
>
> ## Q10: MathVista results
> ## A10:
> The performance drop on MathVista is due to using a larger cache refresh interval (32) to maximize throughput, as shown in Table 10. This achieves a 9.9× speedup (Table 3), but leads to a 2.6% accuracy drop from reduced attention fidelity. Multimodal tasks like MathVista are more sensitive to this due to tighter alignment between vision and language. Using a smaller interval like 4 can restore full accuracy (59.2%) with reasonable speed (19.5 tok/s). We see this as a practical trade-off.

---

> > ### Comment · Reviewer_MBqz · 2025-11-27
> > **Thanks for the rebuttal; remaining concerns on hyperparameter sensitivity**
> >
> > I thank the authors for their detailed and comprehensive response.
> >
> > The additional data regarding the FLOPs/memory overhead (A1) and the comparison with Autoregressive models (A7) have effectively addressed my concerns regarding the theoretical efficiency and competitive landscape of Fast-dLLM. The clarification on the distinction from Block Diffusion (A2) is also well-received.
> >
> > However, I have decided to maintain my score.
> >
> > My primary remaining concern lies in the sensitivity and manual tuning of hyperparameters, which affects the robustness of the proposed method in diverse scenarios:
> >
> > Trade-off in Complex Tasks (A10): The response regarding MathVista reveals a significant trade-off. To recover accuracy (from 56.6% to 59.2%), the cache refresh interval had to be drastically reduced (from 32 to 4), which likely diminishes the speedup advantage significantly. This indicates that the "default" settings are not universally robust and that the method is highly sensitive to the complexity of the downstream task.
> >
> > Lack of Adaptive Mechanisms (A8): While the authors state that a 0.9 threshold works generally, the reliance on fixed scalar hyperparameters (threshold, factor, refresh interval) without a proposed adaptive mechanism (e.g., dynamic adjustment based on attention entropy or real-time similarity metrics) limits the "plug-and-play" nature of the work. In a production environment, requiring users to manually tune these parameters to balance the speed/accuracy frontier for different modalities or prompt lengths is a non-trivial drawback.

---

> > > ### Author Response · Authors · 2025-11-27
> > > **Response to the remaining concerns of Reviewer MBqz**
> > >
> > > Thank you again for your thoughtful feedback and for acknowledging our clarifications regarding efficiency and model comparisons. We fully understand your concerns around hyperparameter sensitivity and the lack of adaptive mechanisms, and we would like to further elaborate on these points.
> > >
> > > Regarding MathVista (A10), while reducing the refresh interval from 32 to 4 does slightly impact the peak throughput, the resulting setting still brings substantial speedup over the baseline. Specifically, with refresh interval = 4, we achieve 19.5 tok/s, which is already 6.87× faster than full-step decoding (2.84 tok/s) and 3.51× faster than half-step decoding (5.56 tok/s), as shown in Table 10. Therefore, even in this more conservative configuration, Fast-dLLM offers significant efficiency gains while recovering nearly full model accuracy, making it suitable for more complex, multimodal applications.
> > >
> > > As for adaptive control (A8), we agree this is an important limitation of the current work. While our fixed thresholds (e.g., 0.9) and factors worked robustly across tasks in our experiments, we concur that automated adjustment based on model confidence, attention dynamics, or real-time similarity metrics would improve the method's plug-and-play usability, especially for scaling across modalities and prompt lengths. We view this as a key avenue for future work, and are actively exploring strategies in this direction, including entropy-based or similarity-aware refresh scheduling.
> > >
> > > We appreciate your constructive feedback and see it as instrumental in guiding the next steps of this research.

---

### Official Review · Reviewer_hYMS · 2025-10-31

**Soundness:** 4
**Presentation:** 3
**Contribution:** 4
**Rating:** 8
**Confidence:** 5

**Summary:**

This paper proposes a novel training-free acceleration framework Fast-dLLM, which aims to address two key bottlenecks in Diffusion LLMs: the lack of KV caching and the quality degradation associated with parallel decoding. First, the paper introduces a block-wise approximate KV Cache mechanism tailored for bidirectional diffusion models. This approach is justified by demonstrating the high similarity of KV activations across adjacent inference steps, which enables cache reuse with a negligible performance drop. Second, the paper posits that the quality degradation in parallel decoding stems from the conditional independence assumption, which disrupts critical token dependencies. To mitigate this, it proposes a confidence-aware parallel decoding strategy that selectively decodes only those tokens exceeding a confidence threshold. This approach is shown to reduce dependency violations while maintaining generation quality. The authors validate their method through comprehensive experiments on the LLaDA and Dream models across multiple LLM benchmarks, and results demonstrate that Fast-dLLM achieves significant throughput improvements (up to 27.6x) while incurring negligible degradation in generation quality.

**Strengths:**

1.The paper introduces a novel, training-free framework that tackles two foundational challenges in dLLM inference.

2.The proposed methods are well-justified and supported by solid theory and experiment phenomenon. The approximate KV cache is empirically validated by the high similarity of KV activations in adjacent steps. The parallel decoding strategy is theoretically supported by Theorem 1, which proves the equivalence of greedy parallel and sequential decoding under high-confidence conditions, motivating the threshold-based approach. Furthermore, the identification of these two phenomena is a valuable contribution, offering clear insights and new avenues for solving these challenges in dLLM inference.

3.The paper provides extensive and sufficient experiments. Main results validate the method's effectiveness. A wide range of ablation studies on variables (e.g., cache block size, confidence thresholds, generation lengths) assess scalability and robustness, with clear analysis and interpretation of the results.

4.This paper is writen well and clear. The structure is logical and easy to follow, making the core concepts and contributions readily understandable.

**Weaknesses:**

The experimental evaluation is primarily limited to comparisons against the baseline dLLM pipelines. The paper lacks an comparison to other existing acceleration techniques for Diffusion LLMs.

**Questions:**

1.The paper's appendix C.4 demonstrates that the 'Factor' decoding strategy significantly outperforms the 'Threshold' strategy in terms of throughput, with only a minor accuracy trade-off. However, the main experimental results (e.g., Table 1) are reported using the slower 'Threshold' strategy. Could the authors clarify why the seemingly superior 'Factor' strategy was not used for the main results? Does the 'Factor' strategy have other un-discussed drawbacks that led to this decision?

2.Could the authors comment on the robustness of the approximate KV cache strategy for very long sequence generation? Is there a risk that approximation errors will accumulate over many blocks, leading to a loss of attentional fidelity to the initial prompt?

3.The qualitative comparisons in appendix B are based on a simple arithmetic prompt, could the authors provide more diverse qualitative examples (e.g., generating idioms or logical pairs) where the risk of dependency failure is more prominent?

---

> ### Author Response · Authors · 2025-11-21
> **Response to Reviewer hYMS**
>
> ## Q1: Comparisons with other dLLM baselines
> ## A1:
>
> As shown in the updated experiments, we include comparisons with two recent and concurrent diffusion LLM optimizations—dLLM-Cache and DPad. These methods focus on distinct axes: dLLM-Cache performs content-based adaptive KV caching, while DPad reduces decoding cost via suffix token pruning. In contrast, our method improves performance through temporal KV reuse and confidence-aware decoding. As shown in the table, Fast-dLLM outperforms each individual method in throughput, and when combined with DPad, achieves multiplicative gains (e.g., 53.8× throughput with only minor accuracy drops, GSM8K-512). These results demonstrate not only competitive standalone performance, but also strong complementarity with existing methods, further positioning Fast-dLLM as a flexible and effective solution in the evolving diffusion LLM landscape.
>
> | GSM8K 5-shot  Gen. Length 256             | PrefixCache | Parallel | Fast-dLLM | dLLM-Cache | DPad  | DPad+Parallel | DPad+Fast-dLLM|
> |------------------------------|--------------------------|-----------------------|-----------|------------|-------|---------------|-------------------------------|
> | Accuracy                     | 79.5                     | 79.2                  | 78.5      | 79.2       | 79.1  | 80.5          | 77.6                          |
> | Throughput                   | 21.2                     | 16.5                  | 54.4      | 18.0       | 7.3   | 20.8          | 54.0                          |
>
> | GSM8K 5-shot Gen. Length 512             | PrefixCache | Parallel | Fast-dLLM | dLLM-Cache | DPad  | DPad+Parallel | DPad+Fast-dLLM|
> |------------------------------|--------------------------|-----------------------|-----------|------------|-------|---------------|-------------------------------|
> | Accuracy                     | 77.0                     | 77.6                  | 77.3      | 78.0       | 80.1  | 81.0          | 79.2                          |
> | Throughput                   | 10.4                     | 18.6                  | 35.3      | 9.2        | 7.2   | 21.2          | 53.8                          |
>
>
> ## Q2: Choice between threshold and factor decoding
> ## A2:
> Thank you for the thoughtful question. While the Factor strategy does offer higher throughput, we chose to report Threshold-based decoding in the main results (e.g., Table 1) to prioritize accuracy preservation, which is often critical in real-world deployment. As shown in Appendix C.2, the Factor method trades off 1–2% accuracy for speed, whereas the Threshold strategy maintains performance closer to the baseline. Additionally, it provides a simpler, more interpretable hyperparameter (a global probability threshold). We view the Factor method as a complementary decoding option—useful for scenarios where speed is the priority and slight accuracy loss is acceptable. For completeness, we include both strategies for flexibility and accuracy–efficiency trade-off.
>
> ## Q3: Robustness of the approximate KV cache strategy for very long sequence generation
> ## A3:
> Thank you for the thoughtful question. Our approximate KV cache is designed to be block-wise refreshed during generation—specifically, we clear and recompute the cache for each new decoding block. As a result, there is no accumulation of approximation error across blocks, and attentional fidelity to the prompt is consistently maintained. This design ensures robustness even in very long sequence generation, as the cached context is regularly updated to reflect the most current decoding state.
>
> ## Q4: More qualitative comparisons for generating idioms
> ## A4:
> We thank the reviewer for the suggestion. In the revised manuscript, Appendix B now includes an additional qualitative example involving idiom completion and explanation, which introduces stronger semantic dependencies. Results (Tables 6–8) show that our methods preserve correctness and semantic fidelity across both arithmetic and linguistic tasks.

---

### Official Review · Reviewer_eGsn · 2025-11-01

**Soundness:** 3
**Presentation:** 3
**Contribution:** 3
**Rating:** 8
**Confidence:** 3

**Summary:**

This paper introduces Fast-dLLM, a training-free acceleration framework for diffusion-based LLMs that addresses their slow inference speed compared to AR models. The method incorporates two key innovations: (1) a block-wise approximate KV Cache mechanism tailored for bidirectional attention that enables cache reuse with negligible performance drop, and (2) a confidence-aware parallel decoding strategy that selectively generates multiple tokens simultaneously based on confidence thresholds to preserve generation quality. Experimental results on LLaDA and Dream models across multiple benchmarks demonstrate remarkable throughput improvement with little accuracy loss, effectively closing the performance gap with autoregressive models.

**Strengths:**

1. Novel adaptation of KV Cache to bidirectional diffusion models via block-wise approximation, with insightful analysis showing high cosine similarity between adjacent steps.
2. Theoretical foundation with Theorem 1 proving the equivalence between greedy parallel and sequential decoding under certain conditions.
3. Comprehensive ablation studies covering key hyperparameters (block sizes, thresholds, generation lengths) and evaluation across models and benchmarks.

**Weaknesses:**

The evaluation of models relies on only four benchmarks (GSM8K, MATH, HumanEval, MBPP for LLaDA), which primarily focus on math reasoning and code generation, missing important capability dimensions like commonsense reasoning (HellaSwag), factual knowledge retrieval (TriviaQA), and real-world code generation (LiveCodeBench, BigCodeBench) that would provide a more comprehensive understanding of the method's generalization and potential failure modes across diverse task types.

**Questions:**

1. Why does the throughput plateau when the block size continues to increase in Fig4 ?
2. Is the KV activation similarity pattern largely consistent, or can it be different among layers?

---

> ### Author Response · Authors · 2025-11-21
> **Response to Reviewer eGsn**
>
> ## Q1: More benchmark results
> ## A1:
> Thank you for the suggestion. To complement our math and code benchmarks, we evaluated our method on IFEval, a benchmark specifically designed to assess instruction following capabilities across diverse task types. As shown in the 512-length setting, Fast-dLLM achieves 3.7× speedup (8.2 → 30.0 tokens/s) with minimal accuracy drop (58.0% → 57.5%), demonstrating strong generalization beyond math and code. This result suggests that our approach remains effective for general-purpose instruction-following tasks.
>
> | Benchmark    | Metric | LLaDA Baseline | +Cache        | +Parallel     | +Cache+Parallel (Fast-dLLM) |
> |--------------|------------|----------------|---------------|---------------|-----------------------------|
> | IFEval 256      | Accuracy        | 57.1            | 53.6          | 57.1          | 54.0                        |
> |              | Throughput          | 15.7            | 19.7          | 30.6          | 36.0                        |
> | IFEval 512      | Accuracy        | 58.0            | 57.5          | 57.9          | 57.5                        |
> |              | Throughput          | 8.2             | 12.0          | 22.2          | 30.0                        |
>
> ## Q2: Why does the throughput plateau when the block size continues to increase in Fig4 ?
> ## A2:
> As shown in Fig. 4, throughput increases with larger block sizes initially, but plateaus beyond a certain point (e.g., ≥64). This is because increasing the block size reduces the frequency of cache updates, improving efficiency up to a point. However, once the cache update overhead becomes negligible, further increasing the block size brings diminishing returns.
>
> ## Q3:  Is the KV activation similarity pattern largely consistent, or can it be different among layers?
> ## A3:
> As shown in the figure 10 of the revised paper, adjacent diffusion steps (e.g., step 16 vs. 17) consistently exhibit high cosine similarity across all layers, indicating that our caching mechanism is well-justified for short-term reuse. However, the similarity decreases in deeper layers as step intervals widen (e.g., step 16 vs. 64), suggesting that KV activations evolve more rapidly toward the top of the network. This non-uniformity across layers and steps points to opportunities for improvement. In future work, we plan to explore layer-aware or adaptive caching strategies that more precisely account for per-layer similarity patterns.

---

### Official Review · Reviewer_SE2t · 2025-11-01

**Soundness:** 4
**Presentation:** 4
**Contribution:** 3
**Rating:** 6
**Confidence:** 4

**Summary:**

This paper proposes Fast-dLLM, a training-free inference acceleration framework for diffusion-based large language models (dLLMs). Unlike autoregressive LLMs, diffusion LLMs cannot directly leverage KV caching or efficient parallel decoding due to their bidirectional denoising process. The authors introduce two key techniques to address this:

* Block-wise Approximate KV Cache: exploits temporal similarity between diffusion steps to reuse attention states across steps, enabling cache reuse in bidirectional attention without retraining.

* Confidence-Aware Parallel Decoding: decodes multiple tokens simultaneously when their predicted confidence exceeds a threshold, with theoretical justification showing approximation to sequential decoding under high-confidence conditions.

Experiments on several diffusion LLMs (e.g., LLaDA, LLaDA-V, Dream) across text, code, and multimodal reasoning tasks demonstrate up to 27.6x speedup with negligible accuracy loss.

**Strengths:**

1. Timely and practically relevant: addresses the major efficiency bottleneck of diffusion-based LLMs, a direction gaining interest.

2. Training-free approach: requires no retraining or model modification, making it directly applicable to existing dLLMs.

3. Comprehensive evaluation: covers both text and multimodal reasoning tasks, with consistent gains across benchmarks.

4. Strong empirical results: large acceleration factors (up to 27.6x) with small degradation make the method attractive for deployment.

**Weaknesses:**

1. Applicability to distilled or few-step diffusion LLMs. It remains unclear whether the proposed caching and confidence-aware decoding strategies would remain effective for distilled diffusion LLMs that operate with only a few or even a single denoising step (e.g., dParallel, arXiv:2509.26488; One-Step Diffusion LLM, OpenReview:P7OzWxOUHK). The reviewer acknowledge that these are concurrent works, while such aggressive timestep reduction is becoming a key trend, similar to continuous diffusion distillation in image/video models. Caching-based acceleration mainly benefits multi-step teacher models, but may offer limited or no gain for student variants without hurting the accuracy, restricting practical adoption.

2. Memory overhead. The block-wise KV caching mechanism likely introduces memory costs, especially for long sequences or large models. The paper does not quantify the memory–speed trade-off or report actual VRAM usage, which is important for understanding deployment feasibility.

3. Baseline comparisons. Recent concurrent and closely related works, such as dLLM-Cache (arXiv:2506.06295) and dPad (arXiv:2508.14148), are not compared or discussed. Even if concurrent, a conceptual comparison highlighting methodological similarities and distinctions would help position Fast-dLLM within this rapidly evolving research landscape.

**Questions:**

How sensitive are the results to the cache window size and confidence threshold? Could a learning-based mechanism adaptively set these parameters?

What is the additional memory footprint introduced by storing bidirectional caches compared to AR caching?

Will the proposed caching mechanism be compatible with TensorRT deployment?

Could the proposed methods extend to multimodal diffusion transformers (e.g., text-to-image or video diffusion models)?

---

> ### Author Response · Authors · 2025-11-21
> **Response to Reviewer SE2t (1/2)**
>
> ## Q1: Applicability to distilled or few-step diffusion LLMs.
> ## A1
> Thank you for raising this important point. We found that our confidence-aware decoding strategy remains effective when applied to distilled few-step diffusion models like dParallel, which naturally exhibit higher confidence due to distillation. Using a threshold of 0.5, dParallel combined with our decoding method already achieves large speedups. Moreover, by further integrating our KV cache, we observed an additional 1.84× throughput gain (from 49.9 to 92.0 tokens/s) with a slight accuracy improvement (from 76.5% to 77.1%) on GSM8K (see table above). The reduced number of decoding steps per block in dParallel improves cache reuse fidelity, making our caching design even more precise. As for One-Step Diffusion LLM, while it explores aggressive timestep reduction, its evaluation focuses primarily on perplexity rather than downstream tasks, so we could not include it in our setting. Exploring how our techniques generalize to extreme distillation regimes remains a valuable direction for future work.
>
> | Gen. Length 256 | Baseline  | dParallel+Threshold 0.5 | dParallel+PrefixCache+Threshold 0.5 |
> |---------------------|------------------|-------------------------|---------------------------------------|
> | GSM8K (5-shot) Accuracy      | 79.3             | 76.5                    | 77.1                                  |
> | GSM8K (5-shot) Throughput    | 6.7              | 49.9                    | 92.0                                  |
>
> ## Q2: Memory overhead
> ## A2:
> Thank you for the insightful question. While our block-wise KV caching mechanism introduces changes in memory access patterns, it does not significantly increase memory usage. The test is conducted in A100-80G GPU. For instance, in the 256+256 case, which means prefix length and generation length are equal to 256, DualCache reduces computation by over 16× (7.68T → 0.48T FLOPs), while memory only increases slightly (15.07G → 15.36G). Even at 1024+1024, VRAM remains well within 16.6GB, showing practicality for large-scale input. PrefixCache reuses only the prefix, while DualCache retains all but the current decoding block, providing more effective reuse at minimal extra memory cost. Overall, our method achieves substantial speedup with marginal VRAM overhead, and scales well to long sequences and large models.
>
> |                     | 256 256           | 512 512           | 1024 1024          |
> |---------------------|-------------------|-------------------|--------------------|
> | LLaDA (Baseline)    | 7.68T/15.07G      | 15.36T/15.20G     | 30.71T/15.44G      |
> | PrefixCache         | 3.84T/15.63G      | 7.68T/16.31G      | 15.36T/17.69G      |
> | DualCache           | 0.48T/15.36G      | 0.48T/15.76G      | 0.48T/16.57G       |

---

> ### Author Response · Authors · 2025-11-21
> **Response to Reviewer SE2t (2/2)**
>
> ## Q3: Baseline comparisons.
> ## A3:
> Thank you for highlighting these concurrent works. While dLLM-Cache focuses on content-based adaptive KV caching using value similarity, our block-wise KV cache relies on temporal similarity across adjacent steps, offering a complementary approach. DPad targets a different axis by pruning redundant suffix tokens to reduce attention cost, whereas our method optimizes the decoding process itself via caching and confidence-aware parallel decoding. Notably, both methods are compatible with our approach, and can be combined for multiplicative speedups, as acknowledged in the DPad paper. We see these works as complementary, addressing different bottlenecks in dLLM inference, and appreciate the opportunity to position Fast-dLLM within this evolving landscape.
>
> | GSM8K 5-shot  Gen. Length 256             | PrefixCache | Parallel | Fast-dLLM | dLLM-Cache | DPad  | DPad+Parallel | DPad+Fasr-dLLM |
> |------------------------------|--------------------------|-----------------------|-----------|------------|-------|---------------|-------------------------------|
> | Accuracy                     | 79.5                     | 79.2                  | 78.5      | 79.2       | 79.1  | 80.5          | 77.6                          |
> | Throughput                   | 21.2                     | 16.5                  | 54.4      | 18.0       | 7.3   | 20.8          | 54.0                          |
>
> | GSM8K 5-shot Gen. Length 512             |  PrefixCache | Parallel | Fast-dLLM | dLLM-Cache | DPad  | DPad+Parallel | DPad+Fasr-dLLM |
> |------------------------------|--------------------------|-----------------------|-----------|------------|-------|---------------|-------------------------------|
> | Accuracy                     | 77.0                     | 77.6                  | 77.3      | 78.0       | 80.1  | 81.0          | 79.2                          |
> | Throughput                   | 10.4                     | 18.6                  | 35.3      | 9.2        | 7.2   | 21.2          | 53.8                          |
>
> ## Q4: Sensitivity of hyperparameters.
> ## A4:
> The sensitivity of our method to the cache block size and confidence threshold is evaluated in Figures 4 and 5. As shown in Figure 4, smaller block sizes (e.g., 4 or 8) slightly improve accuracy but incur cache update overhead, while larger sizes (e.g., 128 or 256) reduce accuracy due to cache mismatch. A moderate block size of 32 achieves the best trade-off, preserving accuracy close to the no-cache setting while delivering a 3.3× throughput gain. This indicates that performance is stable across a broad range, with our chosen setting being robust. Figure 5 shows that higher thresholds improve accuracy (Figure 5a) but reduce the number of tokens decoded per step (Figure 5b). Our selected threshold (0.9) balances these factors well—achieving near-maximum accuracy with reduced computation (Figure 5c). Overall, our dynamic threshold strategy consistently outperforms fixed-token baselines across accuracy and efficiency. While our current setup uses fixed values selected empirically, we agree that a learning-based approach to adapt threshold and block size is a promising direction and leave it to future work.
>
> ## Q5: Further optimization with TensorRT
> ## A5:
> The proposed caching mechanism is conceptually compatible with TensorRT deploymen. While our current implementation is in PyTorch, we are actively integrating the algorithm into efficient inference frameworks such as vLLM and SGLang, which paves the way for deployment in production-ready systems.
>
> ## Q6: Extend to multimodal diffusion transformers
> ## A6:
> Our method can extend to multimodal diffusion transformers such as text-to-image or video models. The core techniques—KV caching and confidence-aware parallel decoding—are architecture-agnostic and applicable to diffusion-based transformers beyond language. For example, Lumina-DiMOO [arXiv:2510.06308] demonstrates a similar approach in multimodal settings with caching and parallel decoding.  Adapting these ideas to spatial or temporal attention in multimodal diffusion models is a promising direction for future work.

---

### Author Response · Authors · 2025-11-27
**General Response to All Reviewers**

We sincerely thank all reviewers for their time and constructive comments (R-SE2t, R-eGsn, R-hYMS, R-MBqz). We are encouraged by the recognition of Fast-dLLM's novelty and effectiveness.

In the individual responses, we have addressed specific questions detailedly. Here, we highlight the major updates and clarifications that cover common concerns:

* Comparison with other Baselines (R-SE2t, R-hYMS): We added comprehensive comparisons with recent diffusion LLM optimization methods (DPad and dLLMCache). Results show Fast-dLLM shows competitive performance among alll the settings and can be combined with these methods for further acceleration.
* Applicability to Distilled Models & New Tasks (R-SE2t, R-eGsn): We demonstrated that our method remains effective on few-step distilled models and extended our evaluation to instruction-following benchmarks, proving the generalizability of Fast-dLLM.
* In-depth Analysis (R-SE2t, R-hYMS, R-eGsn, R-MBqz): We provided further clarifications on memory overhead (minimal impact), robustness for long sequences, and hyperparameter sensitivity, supported by additional data.

We hope these additional experiments and analyses have resolved the concerns raised. We would appreciate it if you could let us know if there are any remaining questions. If our response has adequately addressed your comments, we kindly hope you might consider raising your score.

Best regards,

The Authors

---

### Meta-Review · Area_Chair_FQzz · 2026-01-13

**Summary:**

This paper tackles a key practical bottleneck of diffusion-based LLMs: despite their non-autoregressive promise, open diffusion LLMs are often slow in practice due to missing KV cache support and quality degradation under parallel decoding. The authors propose Fast-dLLM, combining (1) block-wise approximate KV caching (including a bidirectional “DualCache”) and (2) confidence-aware parallel decoding with a simple, theoretically-motivated rule to safely decode multiple tokens when confidence is high. Across LLaDA and Dream, the reported gains are large (up to order-of-magnitude throughput improvements) with small accuracy impact, and the overall approach is training-free and easy to adopt.

The reviewer discussion is largely positive.

**Reviewer Concerns:**

The rebuttal addressed most of the concrete experimental concerns well:

1. the authors added comparisons to concurrent diffusion-LLM acceleration baselines, showed the method remains useful on few-step distilled models,

2. expanded task coverage beyond math/code (e.g., instruction-following),

3. a few more clarifications, including memory overhead with explicit FLOPs/VRAM reporting, and the threshold vs. factor decoding trade-off (accuracy vs. throughput), and why cache refresh does not accumulate error across blocks and added additional qualitative examples, etc..

The main remaining concern is robustness / hyperparameter sensitivity, especially for harder or multimodal tasks where achieving near-baseline accuracy may require more conservative refresh/threshold settings that reduce peak speedups. Reviewers also offered some future improvements that can be in future work.

**Reviewer Scores:**

I think all reviewers likely keep their scores (still positive)

---

### Decision · Program_Chairs · 2026-01-26

Accept (Poster)